



Atmospheric
Chemistry
and Physics

# Multi-model comparison of the volcanic sulfate deposition from the 1815 eruption of Mt. Tambora

**Lauren Marshall**[1], **Anja Schmidt**[1,a], **Matthew Toohey**[2,3], **Ken S. Carslaw**[1], **Graham W. Mann**[1,4], **Michael Sigl**[5], **Myriam Khodri**[6], **Claudia Timmreck**[3], **Davide Zanchettin**[7], **William Ball**[8,9], **Slimane Bekki**[10], **James S. A. Brooke**[11], **Sandip Dhomse**[1], **Colin Johnson**[12], **Jean-Francois Lamarque**[13], **Allegra LeGrande**[14], **Michael J. Mills**[13], **Ulrike Niemeier**[3], **James O. Pope**[15], **Virginie Poulain**[6], **Alan Robock**[16], **Eugene Rozanov**[8,9], **Andrea Stenke**[8], **Timofei Sukhodolov**[9], **Simone Tilmes**[13], **Kostas Tsigaridis**[14], **and Fiona Tummon**[8]

[1]Institute for Climate and Atmospheric Science, School of Earth and Environment, University of Leeds, UK
[2]GEOMAR Helmholtz Centre for Ocean Research Kiel, Kiel, Germany
[3]Max Planck Institute for Meteorology, Hamburg, Germany
[4]National Centre for Atmospheric Science, University of Leeds, UK
[5]Laboratory of Environmental Chemistry, Paul Scherrer Institut, 5232 Villigen, Switzerland
[6]IRD/IPSL/Laboratoire d'Océanographie et du Climat, Paris, France
[7]Department of Environmental Sciences, Informatics and Statistics, University Ca' Foscari of Venice, Mestre, Italy
[8]Institute for Atmospheric and Climate Science, ETH Zurich, Zurich, Switzerland
[9]PMOD/WRC, Davos, Switzerland
[10]LATMOS-IPSL, Université UPMC/Paris-Sorbonne, Université UVSQ/Paris Saclay, CNRS/INSU, Paris, France
[11]School of Chemistry, University of Leeds, UK
[12]Met Office Hadley Centre, Exeter, UK
[13]Atmospheric Chemistry Observations and Modeling Laboratory, National Center for Atmospheric Research, Boulder, CO, USA
[14]NASA Goddard Institute for Space Studies and Center for Climate Systems Research, Columbia University, New York, NY, USA
[15]British Antarctic Survey, Cambridge, UK
[16]Department of Environmental Sciences, Rutgers University, New Brunswick, NJ, USA
[a]now at: Department of Chemistry, University of Cambridge, UK and Department of Geography, University of Cambridge, UK

**Correspondence:** Lauren Marshall (eelrm@leeds.ac.uk)

**Abstract.** [TS1]The eruption of Mt. Tambora in 1815 was the largest volcanic eruption of the past 500 years. The eruption had significant climatic impacts, leading to the 1816 "year without a summer", and remains a valuable event from which to understand the climatic effects of large stratospheric volcanic sulfur dioxide injections. The eruption also resulted in one of the strongest and most easily identifiable volcanic sulfate signals in polar ice cores, which are widely used to reconstruct the timing and atmospheric sulfate loading of past eruptions. As part of the Model Intercomparison Project on the climatic response to Volcanic forcing (VolMIP), five state-of-the-art global aerosol models simulated this eruption. We analyse both simulated background (no Tambora) and volcanic (with Tambora) sulfate deposition to polar regions and compare to ice core records. The models simulate overall similar patterns of background sulfate deposition, although there are differences in regional details and magnitude. However, the volcanic sulfate deposition varies considerably between the models with differences in timing, spatial pattern and magnitude. Mean simulated deposited sulfate on

Please note the remarks at the end of the manuscript.

Antarctica ranges from 19 to 264 kg km$^{-2}$ and on Greenland from 31 to 194 kg km$^{-2}$, as compared to the mean ice-core-derived estimates of roughly 50 kg km$^{-2}$ for both Greenland and Antarctica. The ratio of the hemispheric atmospheric sulfate aerosol burden after the eruption to the average ice sheet deposited sulfate varies between models by up to a factor of 15. Sources of this inter-model variability include differences in both the formation and the transport of sulfate aerosol. Our results suggest that deriving relationships between sulfate deposited on ice sheets and atmospheric sulfate burdens from model simulations may be associated with greater uncertainties than previously thought.

## 1 Introduction

Mt. Tambora in Indonesia (8.2° S, 118.0° E) erupted in April 1815 (e.g. Oppenheimer, 2003) and had a considerable impact on climate, leading to widespread tropical and Northern Hemisphere (NH) mean cooling of $\sim 1\,°C$ and a "year without a summer" in 1816 (e.g. Raible et al., 2016). Volcanic sulfate aerosol, produced from the oxidation of sulfur dioxide ($SO_2$) emitted into the atmosphere by volcanoes, is transported throughout the atmosphere and deposited to the surface by both wet and dry processes, and some is eventually incorporated into polar ice (e.g. Robock, 2000). Bipolar volcanic sulfate deposition signals are presumed to result from tropical eruptions, whereby sulfur entering the tropical stratosphere is converted to sulfate aerosol, which is transported globally by the Brewer–Dobson circulation (e.g. Trepte et al., 1993; Langway et al., 1995; Robock, 2000; Gao et al., 2007). Polar ice core deposition signals typically start around 0.5–1 year after a large tropical eruption and remain elevated for approximately 2–3 years (Robock and Free, 1995; Sigl et al., 2015). Throughout the last 2500 years, polar ice core records show over 200 sulfate spikes, which have been used to estimate the timing, evolution and magnitude of radiative forcing of climate caused by volcanic eruptions during this period (Sigl et al., 2015). The 1815 eruption of Mt. Tambora produced the sixth largest bipolar sulfate signal of the last 2500 years (Sigl et al., 2015).

Determining the stratospheric aerosol properties of the 1815 Mt. Tambora eruption such as spatial extent of the sulfate aerosol cloud, aerosol optical depth and aerosol size distribution bears substantial uncertainties, which ultimately affects the quantification of its climatic impacts using climate models. As part of the Model Intercomparison Project on the climatic response to volcanic forcing (VolMIP) (Zanchettin et al., 2016), which is a Coupled Model Intercomparison Project Phase 6 (CMIP6) endorsed activity (Eyring et al., 2016), coordinated simulations of the 1815 eruption of Mt. Tambora were performed with five state-of-the-art global aerosol models. Our study, motivated by the uncertainty that remains in the climatic forcing from this eruption, investigates the sources of uncertainty in the sulfate deposition to polar regions in these simulations and discusses implications for reconstructions of historic volcanic forcing.

Previous reconstructions of volcanic sulfate aerosol properties used to force climate models scaled the average sulfate deposited on Antarctica and Greenland to the hemispheric atmospheric sulfate aerosol burden (e.g. Gao et al., 2007; Crowley and Unterman, 2013; Sigl et al., 2015). Scaling factors (ratios of the hemispheric sulfate aerosol burden to the sulfate deposited at the poles) were based on the ratio of these two quantities as observed after the eruption of Mt. Pinatubo in 1991 and from the estimated atmospheric burden and measured deposited radioactive material after nuclear bomb tests. Previous climate model simulations of the ratio between atmospheric sulfate burden and polar-deposited sulfate and were also used to derive the scaling factors (Gao et al., 2007, 2008). These scaling factors may not hold for larger eruptions where volcanic sulfate aerosol particles can grow larger, increasing their sedimentation rate (e.g. Pinto et al., 1989; Timmreck et al., 2009). Toohey et al. (2013) also found that differences in the dynamical response to large-magnitude eruptions changed the spatial distribution of the deposited sulfate. Furthermore, available ice core measurements are not evenly distributed over both ice caps, and large spatial variations in the sulfate deposition fluxes can exist between individual ice cores due to differences in local accumulation rates and sulfate redistribution by snow drift (Clausen and Hammer, 1988; Zielinski et al., 1997; Cole-Dai et al., 1997, 2000; Wolff et al., 2005; Gao et al., 2006, 2007). It is therefore important that a range of ice core records from different geographical regions is used to estimate the average volcanic sulfate deposited on each ice cap. Previous studies using only a few ice cores to reconstruct volcanic forcing histories may be biased (e.g. Zielinski, 1995, 1996; Crowley, 2000), although it has been demonstrated that deposition fluxes derived from single ice cores at high-accumulation sites are representative of total ice sheet deposition (Toohey and Sigl, 2017). Gao et al. (2007), who analysed 44 ice cores to investigate the spatial distribution of volcanic sulfate deposition during the last millennium, found larger average deposited sulfate on Greenland (mean deposition of 59 kg km$^{-2}$, using 22 ice cores) than on Antarctica (mean deposition of 51 kg km$^{-2}$, 17 ice cores) for the eruption of Mt. Tambora. However, Sigl et al. (2015) found, using additional high-temporal-resolution ice core records in Antarctica (Sigl et al., 2014), average Antarctic deposited sulfate of 46 kg km$^{-2}$ and a smaller average deposited sulfate on Greenland of 40 kg km$^{-2}$, with both averages smaller than the averages provided by Gao et al. (2007). Although in Sigl et al. (2015) the Antarctic average was derived with 17 ice core records, the Greenland average was calculated from only 2 ice cores (NEEM and NGRIP) compared to the 22 cores used for Greenland in Gao et al. (2007).

Previous modelling studies that have investigated the sulfate deposition from the 1815 eruption of Mt. Tambora have

**Table 1.** Description of models. Modal vs. sectional aerosol size distributions are described in the text.

| Model | Horizontal resolution | Model top, model levels | Aerosol size distribution | Stratospheric compounds | Het. chem.[a] | OH | Sulfur source species | QBO | Reference |
|---|---|---|---|---|---|---|---|---|---|
| CESM1(WACCM) | $0.94° \times 1.25°$ | $4.5 \times 10^{-6}$ hPa, 70 levels | Modal, 3 modes | Sulfate, PSC, organics | Y | Interactive | OCS (337 pptv), DMS, anthrop[b]. SO$_2$ volcanic[c] SO$_2$ | Nudged | Mills et al. (2016, 2017) |
| MAECHAM5-HAM | $2.8° \times 2.8°$ (T42) | 0.01 hPa, 39 levels | Modal, 7 modes | Sulfate | N | Prescribed | OCS ($\sim$ 500 pptv), DMS | NA | Stier et al. (2005), Niemeier et al. (2009) |
| SOCOL-AER | $2.8° \times 2.8°$ (T42) | 0.01 hPa, 39 levels | Sectional, 40 size bins | Sulfate, PSC | Y | Interactive | OCS (337 pptv), DMS, CS2, anthrop. SO$_2$, volcanic SO$_2$ | Nudged | Sheng et al. (2015a) |
| UM-UKCA | $1.25° \times 1.875°$ (N96) | 84 km, 85 levels | Modal, 7 modes | Sulfate, PSC, organics, meteoric dust | Y | Interactive | OCS ($\sim$ 500 pptv), DMS, anthrop. SO$_2$, volcanic SO$_2$ | Internally generated | Dhomse et al. (2014), Brooke et al. (2017) |

[a] Heterogeneous chemistry. [b] Preindustrial anthropogenic SO$_2$. [c] Volcanic SO$_2$ indicates SO$_2$ from passively degassing volcanoes.

failed to reproduce the magnitude of the measured deposited sulfate on both ice caps compared to ice core records, although the models were able to capture the spatial pattern (Gao et al., 2007; Toohey et al., 2013). Gao et al. (2007) found the model-simulated mean deposited sulfate to be a factor of 2 greater than the ice-core-derived estimate, with average Antarctic deposited sulfate of 113 kg km$^{-2}$ and smaller Greenland deposited sulfate of 78 kg km$^{-2}$. Toohey et al. (2013), in contrast, found higher deposition to Greenland and, although matching the spatial pattern of deposited sulfate on Antarctica remarkably well, found model-simulated mean deposited sulfate to be $\sim$ 4.7 times greater than inferred from ice cores. Differences between simulated and measured deposited sulfate could be caused by inaccuracies in the model representation of several physical processes such as the formation and transport of sulfate aerosol, sedimentation, cross-tropopause transport and deposition processes (e.g. Hamill et al., 1997; SPARC, 2006). Neither of the models used by Toohey et al. (2013) and Gao et al. (2007) included a representation of the quasi-biennial oscillation (QBO), which may significantly impact the initial aerosol dispersion (e.g. Trepte et al., 1993). Furthermore, uncertainties exist in the source parameters used for simulating the eruption in models such as the SO$_2$ emission magnitude and emission height.

In general, sulfate deposited on the polar ice caps is only a small fraction of the sulfate deposited globally (e.g. Toohey et al., 2013) and there remains uncertainty surrounding the partitioning of the 1815 Mt. Tambora volcanic sulfate aerosol between both hemispheres. Model results can aid in the interpretation of the ice core estimates by allowing us to assess the relationship between the simulated atmospheric sulfate aerosol burdens and the simulated deposited sulfate.

In this paper we focus on the model-simulated sulfate deposition and the implications for reconstructions of historic volcanic forcing by analysing the deposited sulfate simulated by four global aerosol models and comparing to ice core records. Section 2 describes the model simulations and ice core records. In Sect. 3 we assess the sulfate deposition simulated under both background (no Tambora) (Sect. 3.1) and volcanically perturbed (with Tambora) conditions (Sect. 3.2)

**Table 2.** Model parameters used for the Tambora simulations.

| Parameter | Value in this study |
|---|---|
| SO$_2$ emission | 60 Tg SO$_2$ |
| Eruption length | 24 h |
| Eruption date | 1 Apr |
| Latitude | Equator[a] |
| QBO phase | Easterly |
| SO$_2$ injection height | 22–26 km[b] |

[a] SO$_2$ was emitted at 0° N in CESM1(WACCM), MAECHAM5-HAM and UM-UKCA and at 8° S in SOCOL-AER and at a longitude of 118° E. [b] The altitude distribution for the SO$_2$ emission varied slightly between the models: SOCOL-AER's SO$_2$ emission flux was between 22 and 26 km, increasing linearly with height from zero at 22 km to max at 24 km and then decreasing linearly to zero at 26 km. MAECHAM5-HAM injected at a single model level at 30 hPa ($\sim$ 24 km). UM-UKCA and CESM1(WACCM) used a uniform injection between 22 and 26 km but as the models are not on regular grids and their vertical resolutions differ, the distribution of the emission over the model grid boxes cannot be exactly the same. As a result, the injection profiles differed slightly between the models.

and compare the simulated deposited sulfate to ice core measurements. We investigate the relationship between hemispheric atmospheric sulfate burdens and mean ice sheet deposited sulfate in Sect. 3.3 and explore reasons for model differences in Sect. 4. Conclusions are presented in Sect. 5.

## 2 Models and ice core data

### 2.1 Model descriptions

Of the five models that took part in the coordinated simulations of the 1815 eruption of Mt. Tambora (Zanchettin et al., 2016), only four simulated the sulfate deposition and are therefore included in our study. Model details are listed in Table 1. In each model aerosol formation and growth is simulated through parameterizations of nucleation, condensation and coagulation. Three of the four models have modal aerosol schemes in that the aerosol particle size distribution is represented by several log-normal modes. SOCOL-AER has a sectional scheme where the aerosol particle size distribution is represented by 40 discrete size bins. The models

simulate the transport of stratospheric aerosol through sedimentation and large-scale circulation by the Brewer–Dobson circulation. The QBO is simulated by all models except for MAECHAM5-HAM and is either internally generated (UM-UKCA) or nudged (CESM1(WACCM) and SOCOL-AER). In CESM1(WACCM), MAECHAM5-HAM and UM-UKCA dry deposition schemes are resistance-based and wet deposition is parameterized based on model precipitation and convective processes, with aerosol removal calculated via first-order loss processes representing in-cloud and below-cloud scavenging (Stier et al., 2005; Mann et al., 2010; Lamarque et al., 2012; Liu et al., 2012; Bellouin et al., 2013; Kipling et al., 2013). In SOCOL-AER dry deposition is calculated by multiplying concentrations in the lowest model level by fixed values depending on surface cover. Wet deposition in SOCOL-AER is not related to the precipitation in the model and tropospheric wet removal rates are 5-day mean lifetimes for $H_2SO_4$ (Sheng et al., 2015a). Apart from MAECHAM5-HAM, the models include interactive hydroxyl radical (OH) chemistry, allowing OH concentrations to evolve throughout the simulations (Sect. 4.1.1). Photolysis rates are not impacted by sulfate aerosol in any of the models.

All four models simulate the 1991 eruption of Mt. Pinatubo in reasonable agreement with observations of the sulfate burden, aerosol optical depth and stratospheric heating (Niemeier et al., 2009; Toohey et al., 2011 TS2; Dhomse et al., 2014; Sheng et al., 2015b; Mills et al., 2016), giving confidence in the models' overall abilities to accurately simulate the atmospheric and climatic effects of a large-magnitude eruption. However, the models vary in the details regarding the model–observation comparisons. For example, MAECHAM5-HAM (Niemeier et al., 2009) and SOCOL-AER (Sheng et al., 2015b) simulated a too-rapid aerosol decay and UM-UKCA (Dhomse et al., 2014) had a low bias in the model-simulated aerosol effective radius compared to observations. Possible reasons for these differences include omitted or underrepresented influences from meteoric particles, too large sedimentation and cross-tropopause transport and too-fast transport from tropics to high latitudes. Conversely, the models differ in the amount of emitted $SO_2$ required to achieve good comparisons to observations with the mass of $SO_2$ emitted by the four models ranging from 10 Tg for UM-UKCA (Dhomse et al., 2014) and CESM1(WACCM) (Mills et al., 2016, 2017) to 12–14 Tg for SOCOL-AER (Sheng et al., 2015b) to 17 Tg for MAECHAM5-HAM (Niemeier et al., 2009; Toohey et al., 2011). For this reason, the use of a common protocol in this study (Sect. 2.2) enables us to better attribute potential differences in the results to model processes rather than to the eruption source parameters.

## 2.2 Experiment setup

The parameters used for the Mt. Tambora simulations are listed in Table 2. Each model simulated the eruption by emitting 60 Tg of $SO_2$ at the approximate location of Mt. Tambora between approximately 22 and 26 km (see details in Table 2 regarding the injection details for each model) and during the easterly QBO phase. This $SO_2$ emission estimate is based on both petrological and ice core estimates (Self et al., 2004; Gao et al., 2008), but there remains uncertainty regarding the amount of $SO_2$ emitted, which could range between $\sim 30$ and $80$ Tg $SO_2$ (e.g. Stoffel et al., 2015). Nevertheless, 60 Tg $SO_2$ remains our best estimate. There is also uncertainty in the altitude of the emission and QBO phase due to the lack of observations. Therefore, the injection altitude and QBO phase were chosen to match those of the 1991 Mt. Pinatubo eruption based on satellite and lidar observations (McCormick and Veiga, 1992; Read et al., 1993; Herzog and Graf, 2010). The eruption was simulated by emitting the $SO_2$ over 24 h on 1 April.

In MAECHAM5-HAM, SOCOL-AER and UM-UKCA simulations were atmosphere-only with prescribed preindustrial sea surface temperatures. In CESM1(WACCM) the simulations were run in a preindustrial coupled atmosphere–ocean mode. Climatological preindustrial settings were used for greenhouse gas concentrations, tropospheric aerosols and ozone as defined by each modelling group. The simulations were run for 5 years and included five ensemble members, except for CESM1(WACCM), which had three members only. The models include additional species and processes compared to earlier modelling studies of Mt. Tambora (e.g. Gao et al., 2007; Toohey et al., 2013). UM-UKCA for example includes meteoric smoke particles (Brooke et al., 2017) and an internally generated QBO. Model output is in the form of monthly means.

## 2.3 Ice core data

The ice cores used in this analysis are provided in Tables S1 and S2 in the Supplement. The Antarctic ice cores are the most extensive array of annually resolved cores that have been used to reconstruct historic volcanic forcing (Sigl et al., 2014, 2015). Greenland ice core records have been compiled from several studies (Table S1). Further ice core measurements of the natural background sulfate deposition fluxes were taken from Lamarque et al. (2013).

Sulfate deposition fluxes are derived from ice cores by multiplying measured sulfate concentrations by the annual ice accumulation rate. To derive the volcanic sulfate deposition flux contribution the natural sulfate background level (e.g. due to marine biogenic sulfur emissions) is calculated in each ice core (the non-volcanic contribution) and a threshold flux value is chosen, above which sulfate is assumed to be of volcanic origin. The ice-core-derived volcanic sulfate deposition flux is then calculated as the difference between a year with the volcanic contribution and the mean of the non-volcanic years, and the resulting reported volcanic sulfate deposition flux is the sum of the fluxes in these perturbed years (Ferris et al., 2011; Cole-Dai et al., 2013; Sigl

Atmos. Chem. Phys., 18, 1–21, 2018 www.atmos-chem-phys.net/18/1/2018/

et al., 2013). Our comparable model-simulated volcanic deposition flux is calculated as the sum of the sulfate deposition anomaly (perturbed run minus control run) over the duration of the deposition signal (~ 2–4 years). For SOCOL, which has a sectional aerosol scheme, diagnostics are available for the wet and dry components of the sulfate deposition. For modal models, each of these components is further split into the contribution from each aerosol size mode simulated in the models (nucleation, Aitken, accumulation, coarse). The sulfate deposition flux is calculated (comparable to the ice-core-derived values) as the sum of all of these wet and dry components with a composition of $SO_4^{2-}$ only. In the following sections, we define "total deposition" as referring to the sum of wet and dry deposition fluxes. We define "volcanic sulfate deposition" to specify the sulfate deposition flux anomaly due to the eruption of Mt. Tambora and use "cumulative deposited sulfate" to specify the time-integrated volcanic sulfate deposition fluxes.

To compare the model-simulated results with ice core values, we calculate two statistical metrics: the normalized mean bias (NMB) and correlation coefficient ($r$). NMB is defined by

$$\text{NMB} = \frac{\sum_{i=1}^{N}(M_i - O_i)}{\sum_{i=1}^{N}(O_i)}, \tag{1}$$

where $O_i$ is the ice-core-derived sulfate deposition and $M_i$ is the simulated sulfate deposition in the model grid box containing the ice core. $N$ is the number of ice core records. For both NMB and $r$, each ice core is given equal weighting. We define a high correlation as $r > 0.7$ and low correlation as $r < 0.3$.

## 3   Results

### 3.1   Preindustrial background sulfate deposition

Figure 1 shows the average annual sulfate deposition fluxes in the preindustrial control simulations (no Tambora) for each model. Areas of high background sulfate deposition fluxes are found in close proximity to sulfur emission sources such as continuously degassing volcanoes (e.g. in South America and Indonesia) and along and near midlatitude storm tracks (30–60°) where aerosol is removed effectively by precipitation (except SOCOL-AER, where the deposition is not affected by precipitation). Continuously degassing volcanic emissions are not included in MAECHAM5-HAM. Sulfate deposition fluxes are higher over the oceans than over the land, mainly due to the emission of marine dimethyl sulfide (DMS). In general, Fig. 1 shows that the models have similar background sulfate deposition patterns, with the global mean total (wet + dry) sulfate deposition flux ranging from $78 \, \text{kg} \, SO_4 \, \text{km}^{-2} \, \text{yr}^{-1}$ (CESM1(WACCM)) to $173 \, \text{kg} \, SO_4 \, \text{km}^{-2} \, \text{yr}^{-1}$ (UM-UKCA).

We find that the preindustrial background global mean atmospheric sulfate burdens are similar between CESM1(WACCM), MAECHAM5-HAM and SOCOL-AER but ~ 2–3 times larger in UM-UKCA (Fig. S1 in the Supplement). Sulfur source species included in each model are listed in Table 1. Although the models have similar background sulfate deposition patterns, the partitioning of wet and dry deposition fluxes differs markedly between the models (Fig. 1, Table 3). MAECHAM5-HAM deposits very little sulfate by dry processes compared to the other models with annual global total dry deposited sulfate a factor of 40 less than the global total wet deposited sulfate. In SOCOL-AER, dry deposited sulfate is approximately half the magnitude of wet deposited sulfate.

The sulfate deposited on Antarctica and Greenland is a very small fraction (less than 1 %) of the sulfate deposited globally. In UM-UKCA the sulfate deposited on the polar ice sheets is dominated by dry deposition, which is supported by observations (Legrand and Mayewski, 1997), especially in the Antarctic interior (Wolff, 2012). In contrast, in MAECHAM5-HAM, SOCOL-AER and CESM1(WACCM) the sulfate deposited on the polar ice sheets is dominated by wet deposition, suggesting an issue with the deposition or precipitation representation. However, we find that the simulated total precipitation compares well between models both globally and over the poles (Figs. S2 and S3) indicating the differences in wet and dry deposition partitioning are due to each model's deposition schemes.

The annual global total deposition for both $SO_2$ and $SO_4$ is listed in Table 3 for each model. Included for reference is the equivalent preindustrial $SO_X$ ($SO_2 + SO_4$) deposition from the multi-model mean of the Atmospheric Chemistry and Climate Model Intercomparison Project (ACCMIP) (Lamarque et al., 2013, their Table S4a). The ACCMIP simulations were set up as time-slice experiments and the multi-model mean listed is an average of six models. UM-UKCA compares well to the ACCMIP multi-model mean for dry $SO_X$, but the wet $SO_X$ is $7 \, \text{Tg} \, \text{S} \, \text{yr}^{-1}$ higher and the $SO_4$ deposition ($29 \, \text{Tg} \, \text{S} \, \text{yr}^{-1}$) is also much higher when compared to the other models ($13–19 \, \text{Tg} \, \text{S} \, \text{yr}^{-1}$). MAECHAM5-HAM has a similar total for wet $SO_X$ compared to the ACCMIP multi-model mean, but dry deposition is a factor of 4 lower. CESM1(WACCM) has a similar total for wet $SO_X$ deposition compared to the ACCMIP multi-model mean but total $SO_X$ is $5 \, \text{Tg} \, \text{S} \, \text{yr}^{-1}$ lower. SOCOL-AER simulates the highest dry $SO_X$ ($18 \, \text{Tg} \, \text{S} \, \text{yr}^{-1}$) and total $SO_X$ ($44 \, \text{Tg} \, \text{S} \, \text{yr}^{-1}$) with total $SO_X$ $10 \, \text{Tg} \, \text{S} \, \text{yr}^{-1}$ greater than the ACCMIP multi-model mean.

Following the analysis of Lamarque et al. (2013) we have taken the average sulfate deposition fluxes from 1850 to 1860 (a non-volcanic period) in several ice cores from Antarctica and Greenland and compared the ice core fluxes to the modelled polar sulfate deposition fluxes in the control simulations (Fig. 2). Ice core meta-data are included in the Supplement (Table S2).

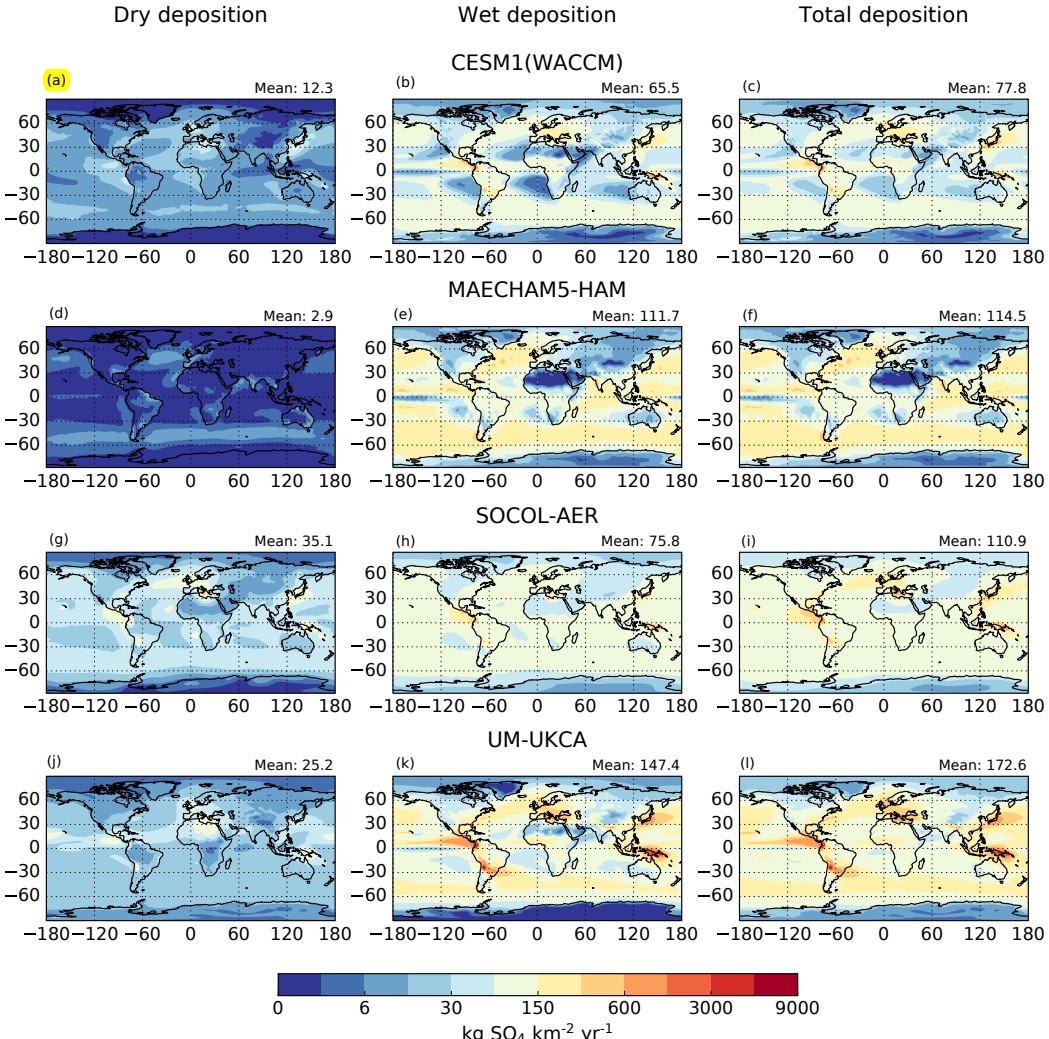

**Figure 1.** Preindustrial background annual dry, wet and total (wet + dry) sulfate deposition fluxes ($kg\,SO_4\,km^{-2}\,yr^{-1}$) (left to right) for each model (top to bottom). The value shown in the top right-hand corner of each plot refers to the global mean sulfate deposition flux. Background fluxes are averages of the annual deposition from five control simulations each with 4 years of data for UM-UKCA, three controls each with 5 years of data for CESM1(WACCM) and one control with 5 years of data for MAECHAM5-HAM and SOCOL-AER. TS3

**Table 3.** Annual global total deposition fluxes of $SO_2$, $SO_4$ and $SO_X$ ($SO_2 + SO_4$) for dry deposition, wet deposition and total dry + wet ($Tg\,S\,yr^{-1}$) in the preindustrial controls and in the ACCMIP multi-model mean (see text). TS4

| Model | Dry SO$_2$ | Wet SO$_2$ | Total SO$_2$ | Dry SO$_4$ | Wet SO$_4$ | Total SO$_4$ | Dry SO$_X$ | Wet SO$_X$ | Total SO$_X$ |
|---|---|---|---|---|---|---|---|---|---|
| CESM1(WACCM) | 5 | 11 | **16** | 2 | 11 | **13** | 7 | 22 | **29** |
| MAECHAM5-HAM | 2 | 2 | **4** | 0.5 | 19 | **19** | 3 | 21 | **24** |
| SOCOL-AER | 12 | 13 | **25** | 6 | 13 | **19** | 18 | 26 | **44** |
| UM-UKCA | 7 | 5 | **12** | 4 | 25 | **29** | 11 | 30 | **41** |
| ACCMIP multi-model mean | – | – | **–** | – | – | **–** | 11 | 23 | **34** |

Overall, CESM1(WACCM), MAECHAM5-HAM and UM-UKCA simulate similar background polar sulfate deposition patterns and magnitudes and compare well to preindustrial ice core sulfate fluxes. Scatter plots of the ice core fluxes vs. those simulated by each model are shown in Fig. 3.

SOCOL-AER simulates slightly higher deposition with reduced regional variability compared to the other models (Fig. 2). However, compared to the ice cores, all models capture the lower sulfate deposition in the interior of Antarctica and higher sulfate deposition toward the coast. The mod-

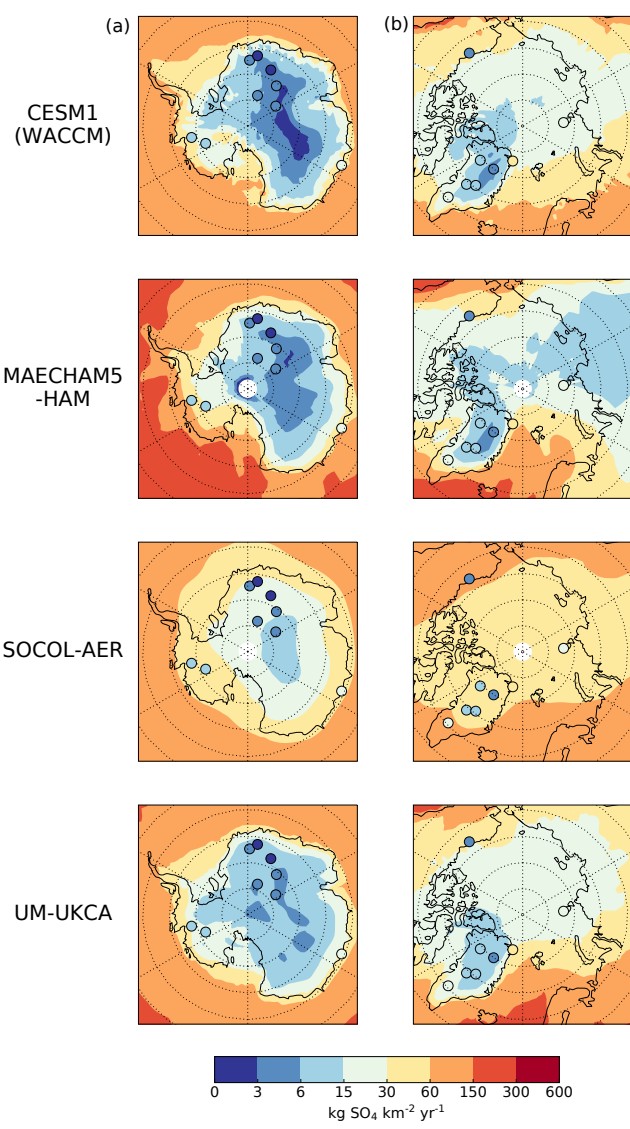

(a)   (b)

CESM1
(WACCM)

MAECHAM5
-HAM

SOCOL-AER

UM-UKCA

0   3   6   15   30   60   150   300   600
kg SO$_4$ km$^{-2}$ yr$^{-1}$

**Figure 2.** Total (wet + dry) sulfate deposition fluxes (kg SO$_4$ km$^{-2}$ yr$^{-1}$) for Antarctica (left) and the Arctic (right) for the preindustrial background from the control simulations (shading) compared to preindustrial ice core sulfate fluxes (filled circles), averaged for 1850 to 1860.

els overestimate Antarctic deposition, particularly in West Antarctica. Antarctic NMB (Sect. 2.3, Eq. 1) ranges from 1.3 (UM-UKCA) to 3.9 (SOCOL-AER) but we find that the model-simulated Antarctic sulfate deposition and Antarctic ice core values are highly correlated for all models with $r$ above 0.9 (Fig. 3). Deposition over the Arctic is also well captured, with MAECHAM5-HAM and CESM1(WACCM) slightly underestimating the sulfate deposition fluxes, both with NMB of $-0.1$. UM-UKCA has a very small positive NMB of 0.01 but SOCOL-AER has the highest Arctic deposition with a NMB of 1.7. None of the models capture

the low flux recorded in Alaska as also found by Lamarque et al. (2013).

The background polar sulfate deposition flux is highly correlated with the simulated mean polar precipitation for CESM1(WACCM), MAECHAM5-HAM and UM-UKCA. Correlation coefficients in the Arctic (60–90°) are between 0.8 (MAECHAM5-HAM and UM-UKCA) and 0.9 (CESM1(WACCM)). The correlation coefficients are slightly higher in the Antarctic ($-60°$ to $-90°$) with $r = 0.9$ for all models. In SOCOL-AER, the higher NMB between simulated polar sulfate deposition fluxes and ice core values is due to the more simplified deposition scheme in this model, which is not connected to the model's simulated precipitation. We find that the Antarctic precipitation in each model matches measured accumulation rates in ice cores (Fig. S3) and with a high correlation with $r$ values of between 0.7 (SOCOL-AER, included for reference) to 0.9 (CESM1(WACCM), UM-UKCA). UM-UKCA and CESM1(WACCM) have very small NMB of $\sim 0.1$. MAECHAM5-HAM has a slightly higher NMB of 0.6 and SOCOL-AER a NMB of 0.8. In the Arctic, the models also capture the precipitation reasonably well compared to the accumulation in the ice cores, with NMB of between 0.1 (UM-UKCA) and 0.5 (CESM1(WACCM)) but low correlation coefficients ($r$ lies between 0.1 and 0.2 for all models). Thus, compared to the ice cores the models capture the magnitude and spatial pattern of the background polar precipitation. Overall, the magnitude of the deposited sulfate in CESM1(WACCM) and MAECHAM5-HAM, where deposition to the ice sheets is dominated by wet deposition, is expected to be driven by the snow accumulation rates across the ice sheets, which are well represented by all models (Fig. S3). In UM-UKCA, although the polar deposition is correlated with the polar precipitation, the ice sheet sulfate deposition mostly occurs by dry deposition. This is because this model deactivates nucleation scavenging when more than a threshold fraction of the cloud water is present as ice, greatly reducing the aerosol scavenging in polar regions. In SOCOL-AER, fewer regional details are captured since the deposition scheme is simpler and is not connected to precipitation, and therefore the deposition mostly reflects the tropospheric distribution of sulfate.

In summary the models simulate similar overall patterns of background sulfate deposition fluxes, although there are differences in the regional details and magnitude. The similarities and realistic deposition patterns amongst the models suggests that the background sulfate emissions, transport and deposition processes are reasonably parameterized. Although SOCOL-AER is less able to simulate regional details, its simplified deposition scheme is still sufficient for the analysis of interhemispheric differences and the temporal evolution of deposition.

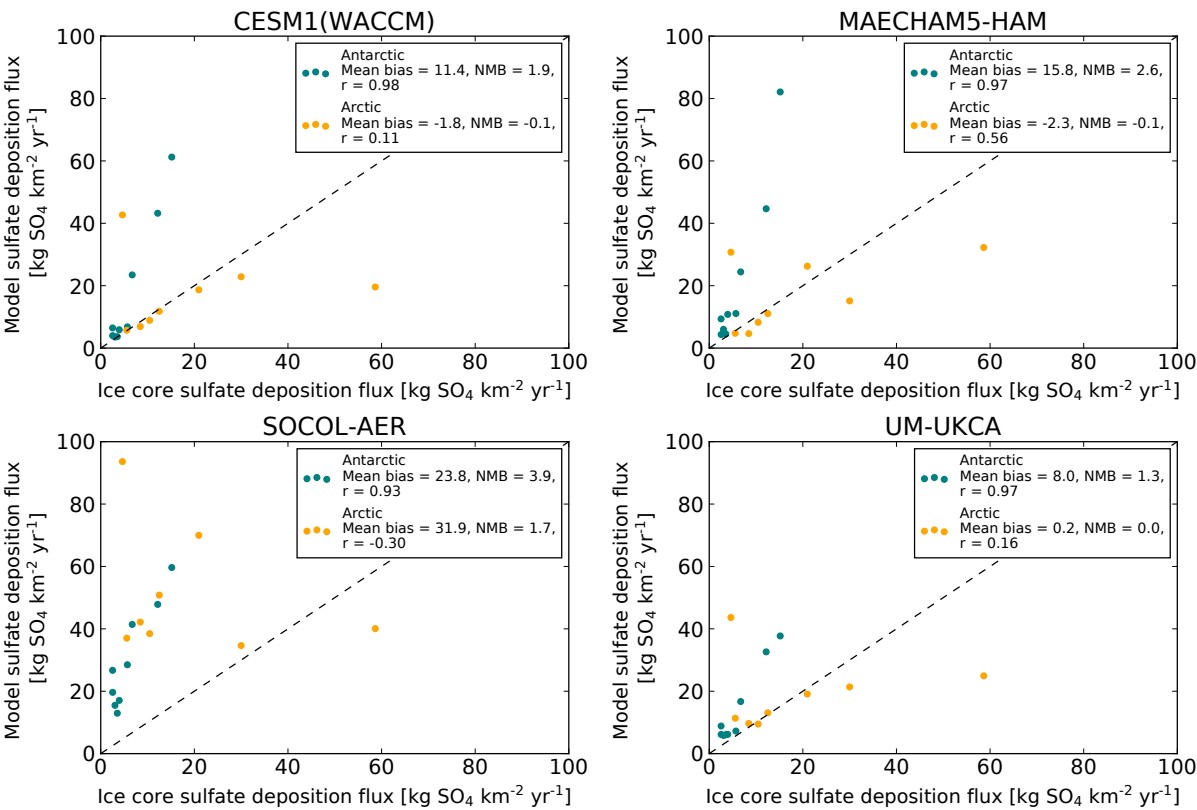

**Figure 3.** Scatter plots of preindustrial background ice core sulfate deposition fluxes vs. simulated preindustrial sulfate fluxes $(\text{kg SO}_4 \text{ km}^{-2} \text{ yr}^{-1})$ in the Antarctic (teal points) and in the Arctic (orange points) for each model. Simulated values represent the grid box value where each ice core is located. The dashed line marks the 1 : 1 line. Included in the legends are the mean bias, normalized mean bias (NMB) and the correlation coefficient ($r$) for the Antarctic and Arctic.

## 3.2 1815 Mt. Tambora eruption sulfate deposition

### 3.2.1 Global sulfate deposition

Figure 4 shows the zonal mean monthly volcanic sulfate deposition (a) and cumulative deposited sulfate (b) simulated by each model and highlights inter-model differences in the timing and spatial distribution of the deposited sulfate. Deposition occurs rapidly in MAECHAM5-HAM with 35 % of the global total deposition occurring in 1815 and the majority (60 %) occurring in 1816. SOCOL-AER simulates the sulfate deposition starting slightly later than in MAECHAM5-HAM, with the majority of the deposition (75 %) occurring in 1816. In contrast, only 9 % of deposition in UM-UKCA occurs in 1815, with 55 % in 1816 and 29 % in 1817. In CESM1(WACCM) the deposition occurs even later, with no deposition occurring in 1815. Instead, 32 % is deposited in 1816, 46 % in 1817 and 17 % in 1818. Deposition is longest in duration in CESM1(WACCM) and global total sulfate deposition remains elevated at the end of the simulation (Fig. 5). In MAECHAM5-HAM deposition returns to near background levels by ∼ 30 months after the eruption and ∼ 40 months for UM-UKCA and SOCOL-AER. We find in-

**Table 4.** Global total cumulative deposited sulfate (Tg S) from dry and wet processes for each model (ensemble mean).

| Model | Dry deposition | Wet deposition | Total deposition |
|---|---|---|---|
| CESM1(WACCM) | 2.4 | 25.4 | 27.8 |
| MAECHAM5-HAM | 0.2 | 28.7 | 28.9 |
| SOCOL-AER | 1.0 | 28.5 | 29.5 |
| UM-UKCA | 3.7 | 25.4 | 29.1 |

dividual ensemble members are similar for each model and the ensemble spread in the global total volcanic sulfate deposition over time is small, as shown in Fig. 5.

In UM-UKCA and CESM1(WACCM) most of the volcanic sulfate is deposited at midlatitudes (30–60°). This contrasts with MAECHAM, where the deposition is globally more uniform, with greater deposition in the polar regions and high deposited sulfate exceeding 360 kg SO$_4$ km$^{-2}$ over West Antarctica, which is completely absent in the other models. In SOCOL-AER, deposition is greatest in the Southern Hemisphere (SH) midlatitudes.

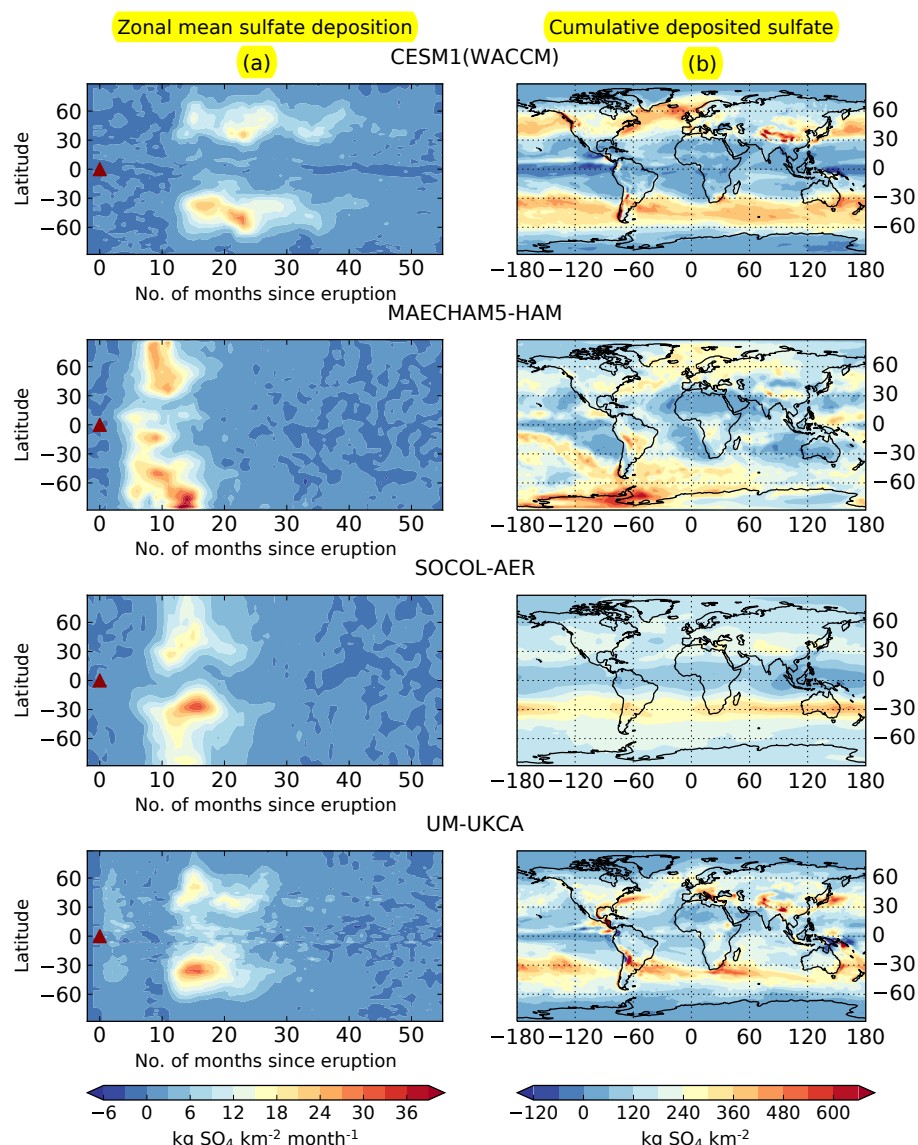

**Figure 4.** Zonal mean volcanic sulfate deposition ($kg\,SO_4\,km^{-2}\,month^{-1}$) (**a**) and cumulative deposited sulfate ($kg\,SO_4\,km^{-2}$) (**b**) for each model (ensemble mean). The red triangle marks the start of the eruption (1 April 1815). Volcanic sulfate deposition is calculated as the difference in total sulfate deposition (wet + dry) between the perturbed and control simulations and this anomaly is summed over the $\sim 5$ years of simulation to produce the cumulative sulfate deposition maps (right column).

The models vary in the simulated relative contribution of wet deposition of sulfate and dry deposition of sulfate to the global total cumulative deposited sulfate (Table 4), although the global total is always dominated by wet deposition, as was also the case with the background sulfate deposition (Fig. 1, Table 3). Dry deposited sulfate in MAECHAM5-HAM is a factor of 15 lower than the dry deposited sulfate simulated by UM-UKCA and CESM1(WACCM). SOCOL-AER also simulates fairly low dry deposited sulfate (1.0 Tg S).

### 3.2.2 Ice sheet sulfate deposition

Although the models simulated similar preindustrial background (no Tambora) polar sulfate deposition (with the exception of SOCOL-AER) (Fig. 2), the simulated polar volcanic sulfate deposition varies in time, space and magnitude between the models. Figure 6 shows the simulated cumulative deposited sulfate for each model compared to the cumulative deposited sulfate measured in ice cores from Greenland and Antarctica for the 1815 Mt. Tambora eruption.

In general, the ice cores from Antarctica show lower volcanic sulfate deposition in East Antarctica and higher de-

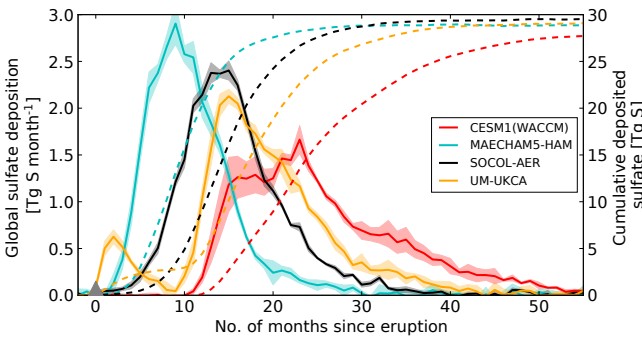

**Figure 5.** Global total volcanic sulfate deposition (Tg S month$^{-1}$) (solid lines – left axis) and global total cumulative deposited sulfate (Tg S) (dashed lines – right axis) for each model (colours). Ensemble mean is shown by the solid line; shading marks 1 SD. The grey triangle marks the start of the eruption (1 April 1815).

position over the Antarctic Peninsula, with deposited sulfate ranging from 13 kg SO$_4$ km$^{-2}$ (East Antarctica, core NUS07-7) to 133 kg SO$_4$ km$^{-2}$ (Antarctic Peninsula, core Siple Dome CE1). In Greenland the ice core estimates range from 25 kg SO$_4$ km$^{-2}$ (B20) to 85 kg SO$_4$ km$^{-2}$ (D3).

We find MAECHAM5-HAM and SOCOL-AER simulate too much deposited sulfate on Antarctica and Greenland compared to the ice cores records (also seen in Toohey et al., 2013), whereas UM-UKCA and CESM1(WACCM) simulate deposited sulfate much closer to the ice core values (Fig. 6). For Antarctica the NMB are 3.9 for MAECHAM5-HAM, 2.0 for SOCOL-AER, −0.5 for CESM1(WACCM) and −0.7 for UM-UKCA. For Greenland the biases are slightly lower: 2.6 for MAECHAM5-HAM, 1.8 for SOCOL-AER, 0.1 for CESM1(WACCM) and −0.5 for UM-UKCA. However, although MAECHAM5-HAM is the model with the highest bias between the simulated cumulative deposited sulfate and ice core values, we find that the simulated Antarctic cumulative deposited sulfate in MAECHAM5-HAM is highly spatially correlated with the ice core values ($r = 0.8$) and Greenland deposition is moderately correlated ($r = 0.6$). Hence MAECHAM5-HAM captures the spatial pattern of the deposited sulfate, especially in Antarctica, with greater deposition on the Antarctic Peninsula and lower deposition in East Antarctica, but the magnitude of the deposition is ∼ 3.7 times too large. Figure 7 shows the ice core values vs. the model-simulated cumulative deposited sulfate. Correlation coefficients are less than ∼ 0.5 for all models except MAECHAM5-HAM, although these models have lower mean biases. A figure where the simulated deposition in MAECHAM5-HAM has been reduced by a factor of 3 to illustrate the well-captured spatial pattern of deposition is included in Fig. S4 (SOCOL-AER is also included in this figure). Both UM-UKCA and CESM1(WACCM), which are the higher-resolution models, simulate a strong gradient in deposition between the low deposition over land and high deposition over sea and, although they match the magnitude

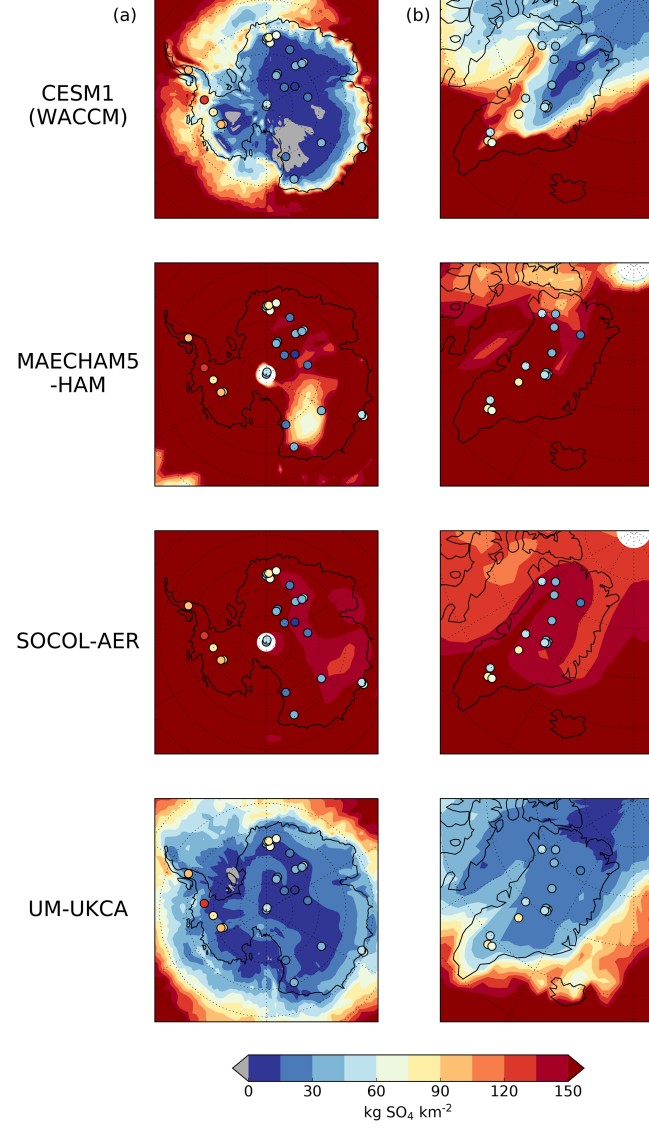

**Figure 6.** Cumulative deposited sulfate (kg SO$_4$ km$^{-2}$) integrated over the whole duration of model simulation (∼ 5 years) on Antarctica (left) and Greenland (right) for each model (ensemble mean). Ice core cumulative deposited sulfate values are plotted as coloured circles. Ice cores from adjacent sites or in close proximity (Table S1) have been slightly relocated to avoid cores completely overlapping. Scaled versions for MAECHAM5-HAM and SOCOL-AER are included in the Supplement (Fig. S4).

of the cumulative deposited sulfate more closely on the ice sheets than SOCOL-AER and MAECHAM5-HAM, they fail to produce the high values of cumulative deposited sulfate on the Antarctic Peninsula.

The polar deposition in UM-UKCA and CESM1(WACCM) more closely follows the models' precipitation field, with correlation coefficients between the polar (60–90°) precipitation (averaged over the 4 years after the eruption) and polar cumulative deposited sulfate (in

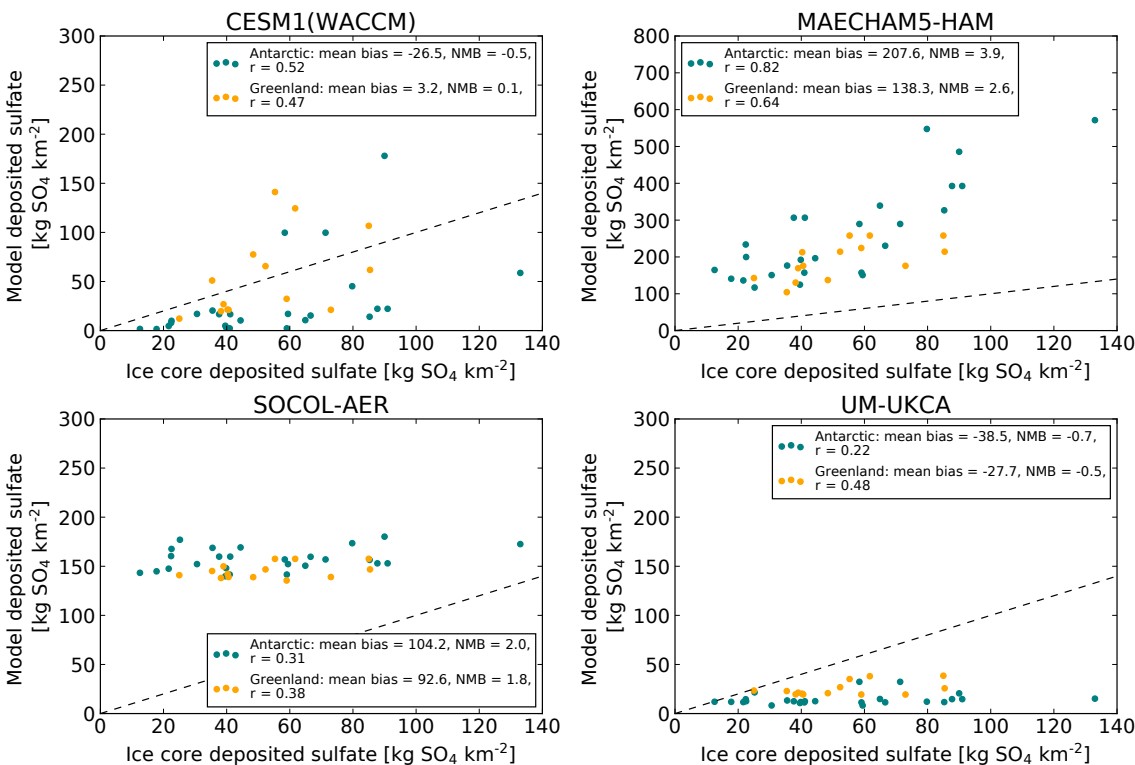

**Figure 7.** Scatter plots of cumulative deposited sulfate ($kg\,SO_4\,km^{-2}$) due to the eruption of Mt. Tambora recorded in ice cores vs. that simulated by each model (ensemble mean) in Antarctica (teal points) and Greenland (orange points). Simulated values represent the grid box value where each ice core is located. The dashed line marks the 1 : 1 line. For each model and for Greenland and Antarctica the mean bias, normalized mean bias (NMB) and correlation coefficient ($r$) between the simulated deposited sulfate and ice core values are shown in the legend. There is an increased $y$-axis scale for MAECHAM5-HAM.

the 4 years after the eruption) of 0.7 and 0.9, respectively. Polar correlation coefficients for MAECHAM5-HAM are 0.6 in the Arctic and 0.4 in the Antarctic. Figure 8 shows the zonal mean precipitation and zonal mean cumulative deposited sulfate in each model. The precipitation in the models is very similar, suggesting that the differences in model-simulated volcanic sulfate deposition arise from differences in the transport of the sulfate aerosol to the polar regions and/or the deposition schemes themselves. The ice sheet sulfate deposition in UM-UKCA remains dominated by dry deposition.

Figure 9 shows for each model the simulated area-mean volcanic sulfate deposition to the Antarctic and Greenland ice sheets over time, compared to two of the highest resolved and most precisely dated ice cores (D4: McConnell et al., 2007 TS5; DIV: Sigl et al., 2014). We find that deposition to both ice sheets peaks first in MAECHAM5-HAM, followed by SOCOL-AER, then UM-UKCA and CESM1(WACCM). The main phase of deposition recorded in the two ice cores falls in time between that simulated by MAECHAM5-HAM and the other models. Compared to DIV and D4, the deposition to the ice sheets in MAECHAM5-HAM is too quick, but too slow in CESM1(WACCM) and UM-UKCA, although the

timing is still relatively well captured for all models. The onset and duration of deposition to the ice sheets simulated by SOCOL-AER is most comparable to the two ice cores, suggesting a good representation of the volcanic aerosol evolution, but simulated deposition is too large (see Fig. 6). The timing of the ice sheet deposition is further explored in Sect. 3.3.

### 3.3 Ice sheet sulfate deposition and relationship to sulfate burdens

The temporal and spatial evolution of the volcanic sulfate deposition ultimately reflects the evolution of the atmospheric volcanic sulfate burdens. Figure 10 shows the zonal mean monthly mean and global total monthly mean atmospheric volcanic sulfate burdens for each model. MAECHAM5-HAM has the fastest conversion of $SO_2$ to sulfate aerosol, with the global peak sulfate burden occurring only 4 months after the eruption (Fig. 10b). This fast conversion is likely due to the lack of interactive OH in the model (Table 1), since OH does not become depleted by reaction with $SO_2$. In UM-UKCA and SOCOL-AER the peak global sulfate burden occurs 6–7 months after the eruption, but the global burden in SOCOL-AER decays more rapidly than in UM-UKCA.

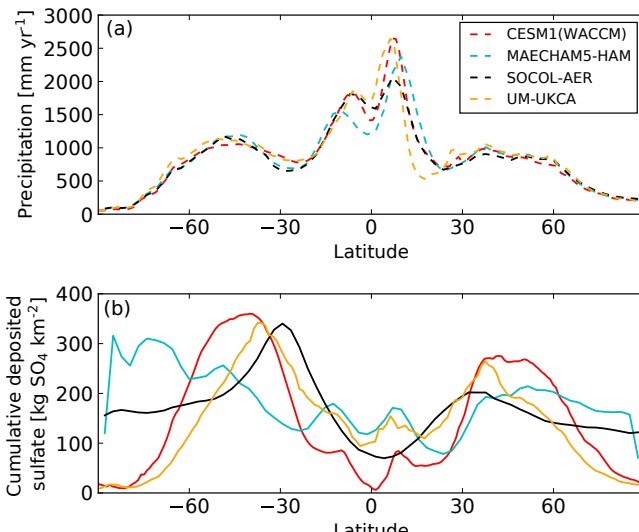

**Figure 8.** Zonal mean precipitation ($\mathrm{mm\,yr^{-1}}$) averaged over the first 4 years after the eruption (top panel, dashed lines) and zonal mean cumulative deposited sulfate ($\mathrm{kg\,SO_4\,km^{-2}}$) in the first 4 years after the eruption (bottom panel, solid lines) for the ensemble mean in each model (colours).

The global burden in CESM1(WACCM) peaks 12 months after the eruption and remains elevated for another 3.5 years (until the end of the simulation) and hence deposition in CESM1(WACCM) is longer lived. The delay in full conversion of $SO_2$ to sulfate aerosol in these models is due to initial depletion of OH, which we explore further in Sect. 4.1.1. In all models there is stronger transport of the sulfate aerosol to the SH compared to the NH (Fig. 10a) due to the Brewer–Dobson circulation, which is stronger in the winter hemisphere.

Here we consider the relationships between the NH sulfate burden vs. the SH sulfate burden, the cumulative sulfate deposited on Antarctica vs. Greenland and, most importantly, between the hemispheric sulfate burdens and the sulfate deposited on each ice sheet.

In all models the SH peak atmospheric sulfate burden is greater than the NH peak atmospheric sulfate burden (Table 5). Ratios between the SH and NH peak burdens are between 1.4 and 1.9. However, despite the larger SH burden, only MAECHAM5-HAM and SOCOL-AER simulate greater Antarctica mean deposited sulfate than in Greenland. CESM1(WACCM) has the smallest deposition ratio (0.3) with mean Greenland deposited sulfate of $109\,\mathrm{kg\,SO_4\,km^{-2}}$ compared to $36\,\mathrm{kg\,SO_4\,km^{-2}}$ in Antarctica. MAECHAM5-HAM and SOCOL-AER have the closest deposition ratio to that derived by Sigl et al. (2015), but with mean deposited sulfate $\sim 4$–6 times larger than the Sigl et al. (2015) estimates. Conversely, and as simulated in UM-UKCA and CESM1(WACCM), the mean deposited sulfate deduced by Gao et al. (2007) for the eruption of Mt. Tambora showed

slightly more mean deposited sulfate on Greenland relative to Antarctica, with a ratio of 0.9, although this ratio is still much larger than in UM-UKCA and CESM1(WACCM). In contrast to MAECHAM5-HAM and SOCOL-AER, where the deposition ratio mirrors the hemispheric split of the sulfate aerosol, deposition ratios for both UM-UKCA and CESM1(WACCM) are dissimilar to the ratio of the hemispheric peak burdens.

Figure 11 shows the simulated deposition to each ice sheet over time as in Fig. 9, except we compare to the hemispheric sulfate burdens. In MAECHAM5-HAM the NH sulfate burden peaks only 2 months after the eruption and the SH burden peaks 4 months after the eruption. The ice sheet deposition follows suit with the majority of deposition to Greenland occurring 8 months after the eruption and peak deposition to Antarctica occurring 14 months after the eruption. However, in the other models the SH burden peaks before the NH burden. The SH burden is greatest between 5 and 7 months after the eruption in these models and the NH burden peaks between 10 and 12 months after the eruption. In contrast to MAECHAM5-HAM, there are no clear separate peaks between the deposition to each ice sheet. In SOCOL-AER both the majority of Greenland and Antarctic deposition occurs between 10 and 20 months after the eruption, which was found to compare well to the timing recorded in two ice cores (Fig. 9). In UM-UKCA and CESM1(WACCM) the main phase of deposition is longer lived and occurs between 10 and 30 months after the eruption. Overall, decay of the atmospheric sulfate burden and deposition to the ice sheets in MAECHAM5-HAM is rapid, occurring within the first 20 months after the eruption, suggesting a fast transport of sulfate aerosol to the poles. We find that in the first $\sim 8$ months after the eruption the sulfate burden in UM-UKCA and CESM1(WACCM) is restricted between $\sim 60^\circ$ S and $\sim 40^\circ$ N (Fig. 10a), with strong gradients in sulfate burden across the SH polar vortex and NH subtropical edge, whereas more sulfate is transported to the poles in MAECHAM5-HAM and SOCOL-AER. Reasons for this are explored in Sect. 4.

Next, we calculate the ratio between the hemispheric peak atmospheric sulfate burdens ($\mathrm{Tg\,SO_4}$) (representing the total amount of sulfate aerosol that has formed) and the average amount of sulfate deposited on each ice sheet ($\mathrm{kg\,SO_4\,km^{-2}}$) for each of the models. We refer to this ratio as the burden-to-deposition (BTD) factor, which is equivalent to the scaling factors derived by Gao et al. (2007) calculated from the observed relationship between the atmospheric burden and deposition of radioactive material after nuclear bomb tests. BTD factors are important for estimating the hemispheric atmospheric sulfate burden and subsequently estimating the forcing of historical volcanic eruptions based on ice core sulfate deposition records (Sect. 1). We calculate the BTD factors for both NH (NH_BTD) and SH (SH_BTD) (Table 6). BTD factors for MAECHAM5-HAM are the same for both the NH and SH, as in Gao et al. (2007), but a fac-

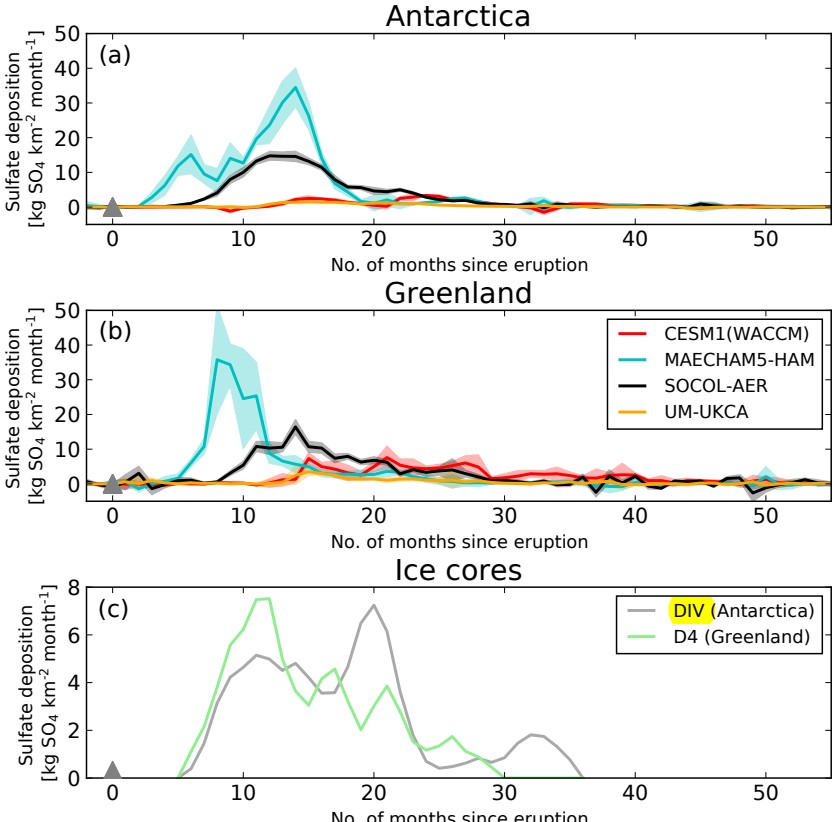

**Figure 9.** Simulated area-mean volcanic sulfate deposition ($kg\,SO_4\,km^{-2}\,month^{-1}$) to the Antarctic ice sheet (top panel) and Greenland ice sheet (middle panel) for each model (colours). Each ice sheet mean is defined by taking an area-weighted mean of the grid boxes in the appropriate regions once a land–sea mask has been applied. Solid lines mark the ensemble mean and shading is 1 SD. In the bottom panel are deposition fluxes from two monthly CE2 resolved ice cores (DIV from Antarctica and D4 from Greenland). The scale is reduced in the bottom panel. The grey triangles mark the start of the eruption.

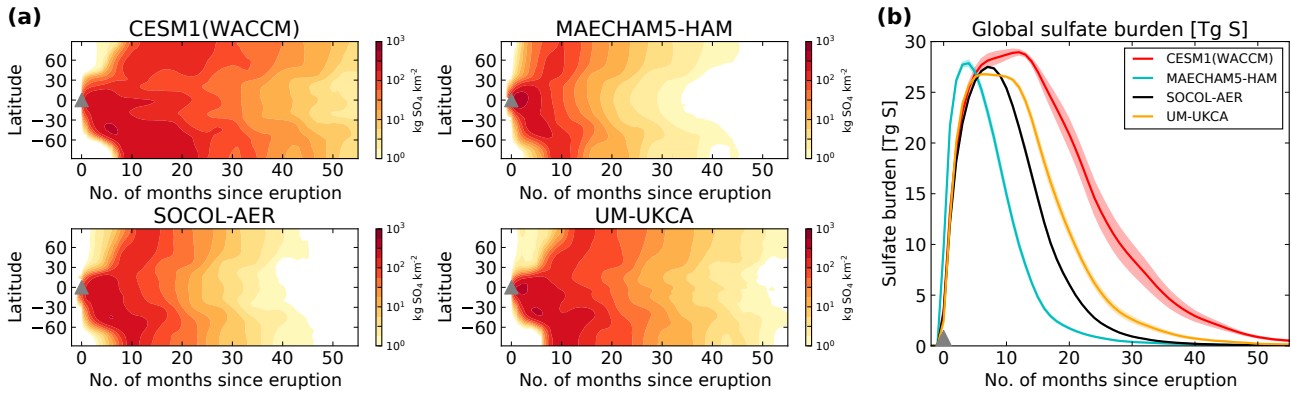

**Figure 10. (a)** Zonal mean atmospheric sulfate burdens in each model ($kg\,SO_4\,km^{-2}$) (ensemble mean) **(b)** global total atmospheric sulfate burdens (Tg S) in each model (colours). Ensemble means are shown by the coloured lines; shadings mark 1 SD. Sulfate burdens are monthly mean anomalies. The grey triangle marks the start of the eruption (1 April 1815).

tor of 5 lower than Gao et al. (2007). CESM1(WACCM), SOCOL-AER and UM-UKCA simulate smaller NH_BTD than SH_BTD, but these factors are different in each model, with the NH_BTD ranging from $0.22 \times 10^9\,km^{-2}$ (SOCOL-AER) to $0.97 \times 10^9\,km^{-2}$ (UM-UKCA) and the SH_BTD from $0.34 \times 10^9\,km^{-2}$ (SOCOL-AER) to $2.91 \times 10^9\,km^{-2}$ (UM-UKCA). All models simulate a NH_BTD less than $1 \times 10^9\,km^{-2}$, but SH_BTD is less than $1 \times 10^9\,km^{-2}$ for

**Table 5.** Greenland and Antarctica ice sheet mean cumulative deposited sulfate and ratio (Antarctica deposition / Greenland deposition) and peak NH and SH sulfate burdens (total atmospheric column burden anomaly) and ratio (SH burden / NH burden) for each model (ensemble mean). Also included is the equivalent mean deposited sulfate on each ice sheet calculated from ice cores (Gao et al., 2007; Sigl et al., 2015). TS6

| Model | Mean Antarctica deposited sulfate (kg $SO_4$ km$^{-2}$) | Mean Greenland deposited sulfate (kg $SO_4$ km$^{-2}$) | Antarctica / Greenland deposition ratio | Peak SH sulfate burden (Tg $SO_4$) | Peak NH sulfate burden (Tg $SO_4$) | SH / NH burden ratio |
|---|---|---|---|---|---|---|
| CESM1(WACCM) | 36 | 109 | 0.3 | 58 | 34 | 1.7 |
| MAECHAM5-HAM | 264 | 194 | 1.4 | 50 | 36 | 1.4 |
| SOCOL-AER | 163 | 148 | 1.1 | 56 | 32 | 1.8 |
| UM-UKCA | 19 | 31 | 0.6 | 56 | 29 | 1.9 |
| Sigl et al. (2015) | **46** | **40** | **1.2** | – | – | – |
| Gao et al. (2007) | **51** | **59** | **0.9** | – | – | – |

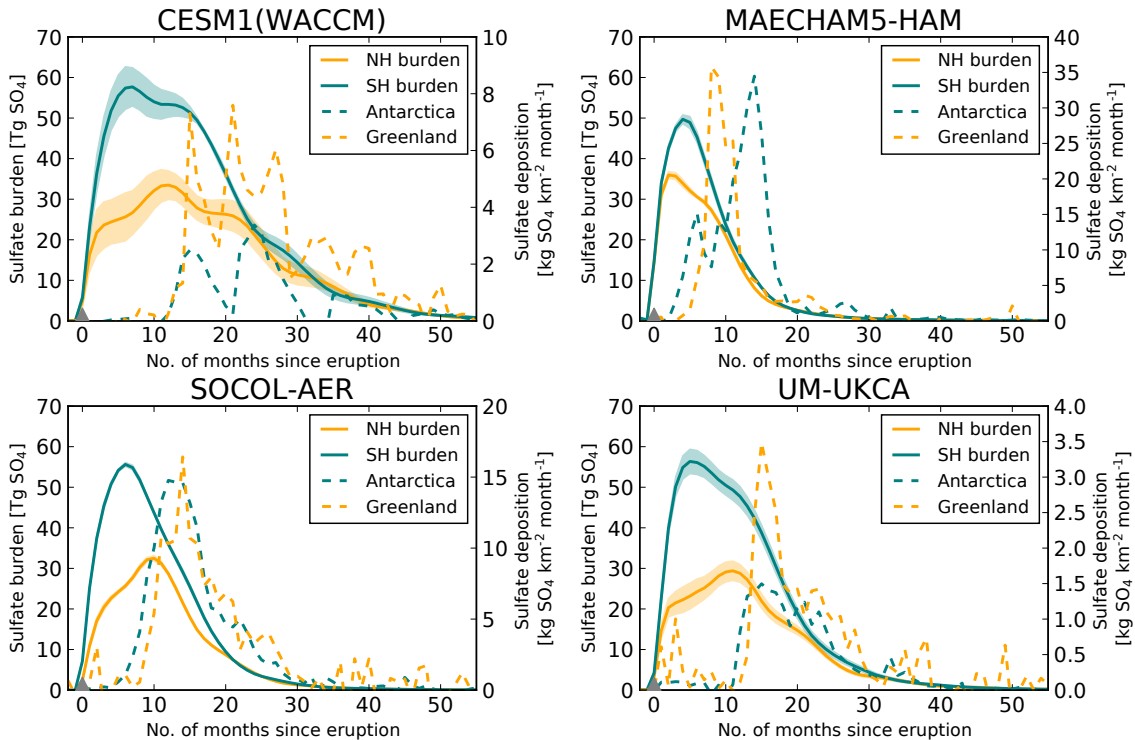

**Figure 11.** Hemispheric atmospheric sulfate burdens (Tg $SO_4$) (solid lines show the ensemble mean and shading is 1 SD) and area-mean ice sheet volcanic sulfate deposition as in Fig. 9 (dashed lines) (kg $SO_4$ km$^{-2}$ month$^{-1}$) for each model. The grey triangles mark the start of the eruption. There are different scales on each secondary *y* axis for ice sheet deposition.

only MAECHAM5-HAM and SOCOL-AER due to the much larger Antarctic deposition in these models compared to UM-UKCA and CESM1(WACCM). The multi-model mean NH_BTD factor is $0.42 \times 10^9$ km$^{-2}$ ($\sim 60\%$ smaller than in Gao et al., 2007) and multi-model mean SH_BTD factor is $1.27 \times 10^9$ km$^{-2}$ ($\sim 30\%$ greater than in Gao et al., 2007). We also find variability in the BTD factors across the individual ensemble members for each model arising due to

internal variability, but ensemble spread is smaller than the inter-model spread.

We also test the sensitivity of the derived model BTD factors in Table 6 when we take polar deposition (60–90° N/S) as opposed to ice sheet deposition, given that both UM-UKCA and CESM1(WACCM) simulate strong gradients in cumulative deposited sulfate across the land–sea boundary (Fig. 6). We find that the BTD factors remain similar for SOCOL-AER and MAECHAM5-HAM but are re-

**Table 6.** Burden-to-deposition (BTD) factors ([a] $10^9$ km$^{-2}$) between the hemispheric peak sulfate burden (Tg SO$_4$) (total atmospheric column burden anomaly) and the mean ice sheet cumulative deposited sulfate (kg SO$_4$ km$^{-2}$) for the four models and from Gao et al. (2007). Included are the values for the ensemble mean factor and the range from individual ensemble members. TS7

| Model | NH_BTD ($10^9$ km$^{-2}$) | | SH_BTD ($10^9$ km$^{-2}$) | |
|---|---|---|---|---|
| | Ensemble mean | Ensemble range | Ensemble mean | Ensemble range |
| CESM1(WACCM) | 0.31 | 0.29–0.34 | 1.63 | 1.44–1.96 |
| MAECHAM5-HAM | 0.19 | 0.14–0.24 | 0.19 | 0.17–0.20 |
| SOCOL-AER | 0.22 | 0.20–0.24 | 0.34 | 0.32–0.35 |
| UM-UKCA | 0.97 | 0.74–1.14 | 2.91 | 2.67–3.30 |
| Multi-model mean | 0.42 | – | 1.27 | – |
| Gao et al. (2007) | **1** | **–** | **1** | **–** |

duced by up to a factor of 3 in UM-UKCA due to the mean polar cumulative deposited sulfate being greater than the mean ice sheet cumulative deposited sulfate (Table S3). In CESM1(WACCM) the SH_BTD is also reduced by a factor of 3, but the NH_BTD remains similar. Overall, the spread in the BTD factors between the models decreases and results in a reduction of the multi-model mean NH_BTD factor from $0.42 \times 10^9$ to $0.28 \times 10^9$ km$^{-2}$ and the SH_BTD from $1.27 \times 10^9$ to $0.54 \times 10^9$ km$^{-2}$.

## 4 Discussion

### 4.1 Differences in deposited sulfate

The spatial pattern and magnitude of deposited sulfate depends on the sources of atmospheric SO$_2$, the transport and mixing of the sulfate aerosol formed throughout the stratosphere and across the tropopause and wet and dry deposition processes (e.g. Hamill et al., 1997; Kremser et al., 2016). In the preindustrial background state (no Tambora) (Fig. 1), all four models examined simulate similar patterns of sulfate deposition, with more sulfate deposited at the midlatitudes and in oceans and near SO$_2$ sources such as continuously degassing volcanoes. In the polar regions, the models also simulate similar sulfate deposition (with the exception of SOCOL-AER) with reasonable comparison to ice core records (Fig. 2). This indicates that the models are realistically simulating aspects of the formation and transport of background sulfate aerosol and subsequent deposition processes.

However, under the volcanically perturbed conditions (with Tambora), the simulated volcanic sulfate deposition differs between all models, with differences in timing, spatial pattern and magnitude. Compared to ice core records of cumulative deposited sulfate for 1815 Mt. Tambora, MAECHAM5-HAM and SOCOL-AER simulate much higher deposition to polar ice sheets, which is $\sim$ 3–5 times greater than the mean ice-core-derived estimates by Gao et al. (2007) and Sigl et al. (2015). UM-UKCA and

CESM1(WACCM) simulate deposition closer in magnitude to the ice core records although in UM-UKCA the sulfate deposited on both ice sheets is $\sim$ 2 times too small compared to the mean ice-core-derived estimates. In CESM1(WACCM) the sulfate deposited on Antarctica is slightly too small but $\sim$ 2 times greater in Greenland compared to the mean ice-core-derived estimates. Considering the models are more comparable in the background state it is likely that the inter-model differences in volcanic deposition are due to differences in the formation of the volcanic aerosol, the stratospheric transport of volcanic aerosol and stratosphere–troposphere exchange, since in the background state most of the deposited sulfate is of tropospheric origin. These processes are discussed in the following sections.

### 4.1.1 Volcanic sulfate formation and transport

The timing and duration of sulfate deposition mirrors that of the atmospheric sulfate burdens. In MAECHAM5-HAM the atmospheric sulfate burden peaks sooner and decays more quickly than in the other models, and ice sheet deposition occurs more rapidly (within the first 2 years after the eruption). The atmospheric sulfate burden in CESM1(WACCM) is still elevated 4 years after the eruption, and hence the deposition signal is also longer lived (Fig. 5). MAECHAM5-HAM is the only model that has prescribed OH (Table 1). OH may become depleted in dense volcanic clouds by reaction with SO$_2$, affecting the rate of sulfate aerosol formation (Bekki, 1995). The background stratospheric OH concentrations are similar between the models (Fig. S5) but in SOCOL-AER, UM-UKCA and CESM1(WACCM), in the first 2 months after the eruption, stratospheric tropical OH becomes depleted, with ensemble mean peak reductions of between 15 and 33 % (Fig. S6). This reduces the rate of sulfate aerosol formation compared to MAECHAM5-HAM, where the SO$_2$ will be more rapidly oxidized, and explains the later peaks in sulfate burdens in these models.

The rapid decay of the sulfate burden in MAECHAM5-HAM also indicates that this model could have faster accumulation of particles and stronger sedimentation compared

to the other models. Although beyond the scope of this paper a more detailed examination of the aerosol microphysical processes and the size of the aerosol particles, on which sedimentation is dependent, will facilitate a greater understanding of some of the model differences identified here.

The high biases in cumulative deposited sulfate in MAECHAM5-HAM and SOCOL-AER compared to ice cores may be caused by a high bias in poleward aerosol transport (e.g. Stenke et al., 2013; Toohey et al., 2013). MAECHAM5-HAM and SOCOL-AER also have the lowest resolution of the four models (Table 1), which may contribute to the high deposition bias since stratospheric circulation and cross-tropopause transport is better represented in higher-resolution models (e.g. Toohey et al., 2013). Gao et al. (2007), using the GISS ModelE, found that simulated deposited sulfate over the poles after the eruption of Mt. Tambora was a factor of 2 too large but that the spatial pattern of deposition recorded in ice cores was well captured. GISS ModelE had a much lower resolution than the models used here ($4° \times 5°$) and a simplified scheme for stratospheric aerosol microphysics. SOCOL-AER and MAECHAM5-HAM have the same dynamical cores and therefore we expect transport to be similar in these models. Hence, the differences in simulated volcanic deposition between SOCOL-AER and MAECHAM5-HAM are likely due to aerosol growth and sedimentation, and the deposition schemes. In UM-UKCA and CESM1(WACCM) the poleward transport of volcanic aerosol may be too weak or midlatitude deposition too strong.

### 4.1.2 Dynamical effects

The direction and strength of the stratospheric winds impacts the transport of sulfate aerosol and hence where it is deposited. UM-UKCA, CESM1(WACCM) and SOCOL-AER have similarly defined QBOs with downward propagating easterly and westerly winds, with the eruption simulated in the easterly phase. MAECHAM5-HAM does not include a QBO and although stratospheric winds were easterly in the MAECHAM5-HAM simulations, we find that these winds are $\sim 20\,\mathrm{m\,s^{-1}}$ weaker than the easterly phase winds in the other models (Fig. S7). This may contribute to the quicker transport and subsequent deposition to the poles in the MAECHAM5-HAM simulations.

In addition to midlatitude tropopause folds, a further location of cross-tropopause transport of sulfate aerosol is the polar winter vortex (e.g. SPARC, 2006; Kremser et al., 2016). The polar vortex inhibits poleward transport (e.g. Shoeberl and Hartmann, 1991), and it has been suggested that variations in the strength of the polar vortex may modulate volcanic aerosol transport and deposition to polar ice sheets (Toohey et al., 2013). We find that the strength of the background climatological winds differs slightly across the models, with the strongest polar jets simulated in CESM1(WACCM) and the weakest in MAECHAM5-

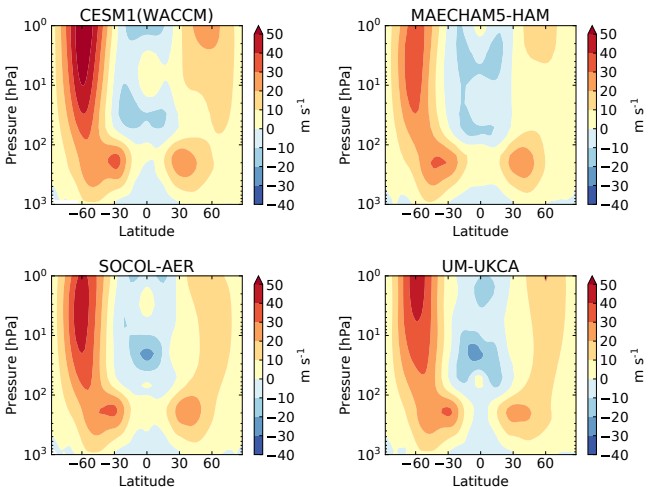

**Figure 12.** Zonal mean zonal wind ($\mathrm{m\,s^{-1}}$) averaged over the first year after the eruption (April 1815–April 1816) in each model simulation (ensemble mean). Zonal wind is output on 36 pressure levels in UM-UKCA, 33 pressure levels in MAECHAM5-HAM and 32 pressure levels in SOCOL-AER. Zonal wind in CESM1(WACCM) is output on an atmosphere hybrid sigma pressure coordinate and has been interpolated to the pressure levels used in UM-UKCA.

HAM (maximum zonal mean zonal winds are $52\,\mathrm{m\,s^{-1}}$ in CESM1(WACCM) and $32\,\mathrm{m\,s^{-1}}$ in MAECHAM5-HAM; Fig. S8). All models simulate a strengthening in the NH and SH polar zonal winds in the first year after the eruption; CESM1(WACCM) simulates the largest zonal mean anomalies and MAECHAM5-HAM the weakest. Figure 12 shows the zonal mean zonal wind averaged over the first year after the eruption in each model. Peak zonal mean SH polar zonal wind is $58\,\mathrm{m\,s^{-1}}$ in CESM1(WACCM), $46\,\mathrm{m\,s^{-1}}$ in SOCOL-AER, $45\,\mathrm{m\,s^{-1}}$ in UM-UKCA and $38\,\mathrm{m\,s^{-1}}$ in MAECHAM5-HAM. Inter-model differences in polar vortex strength may therefore contribute to differences in polar sulfate deposition. Following this hypothesis, the strong SH polar vortex simulated by CESM1(WACCM) may contribute to the lower deposited sulfate on Antarctica in this model and, likewise, the relatively weaker polar vortex in MAECHAM5-HAM may contribute to the greater deposited sulfate on Antarctica. In contrast, UM-UKCA simulates average polar vortex winds but the smallest deposited sulfate on Antarctica. Therefore, it appears to be a combination of factors that drive the inter-model differences in simulated polar volcanic sulfate deposition.

### 4.1.3 Deposition schemes

Differences in the deposition schemes contribute to the inter-model differences. The simplified scheme in SOCOL-AER results in deposition following more closely the atmospheric distribution of sulfate. The differences between wet and dry deposition simulated across the models are due to the individ-

ual deposition scheme parameterizations. The implication of these differences in dictating the resulting total sulfate deposition remains uncertain. However, since inter-model differences in volcanic sulfate deposition patterns appear unrelated to differences between climatological wet and dry deposition patterns, the proportion of wet vs. dry deposition is likely of secondary importance compared to differences between the models in aerosol transport processes including sedimentation and stratosphere–troposphere exchange. Smoother topography in the lower-resolution models will also influence the spatial pattern of deposition.

The realistic deposition of background sulfate suggests that the scavenging and deposition processes in the models are reasonably parameterized and thus that inter-model differences in the Tambora case are more likely due to differences in stratospheric transport and stratosphere–troposphere exchange as described above. However, due to the higher sulfate burdens in the perturbed case, differences in deposition due to the schemes may become more pronounced. Scavenging and deposition parameterizations are highly uncertain, and the chance that such parameterizations become unrealistic under the large sulfate aerosol loadings associated with a Tambora eruption cannot be discounted and should be explored in future work.

## 4.2 Implications for model differences in simulated sulfate deposition

Using just four global aerosol models, we find large differences in the mean deposited sulfate on the Antarctic and Greenland ice sheets. The multi-model mean BTD factors, which relate the atmospheric sulfate burdens to the deposition at the ice sheets, differ from the estimates by Gao et al. (2007) by $\sim 60\%$ in the NH and $\sim 30\%$ in the SH, although the Gao et al. (2007) estimates are within or close to the multi-model spread. We find that the multi-model spread in BTD factors is reduced when we take a polar cap average of deposition as opposed to the average ice sheet deposition because simulated deposition is more similar amongst the models when a greater area average is considered. Due to the large gradient between land and sea deposition simulated in CESM1(WACCM) and UM-UKCA, mean polar deposition does not represent the mean ice sheet deposition and BTD factors are therefore sensitive to the areas chosen to represent the polar–ice sheet deposition. This makes it difficult to estimate accurately the relationship between ice sheet deposition and sulfate aerosol loading in the models. We highlight this to emphasize caution when determining BTD factors in future modelling studies. Furthermore, although these simulations aimed to follow a common protocol, the injection setup did differ between models due to differences in the ways modelling groups interpret and simulate a volcanic injection (Table 2). This is a common problem in multi-model comparisons and makes it more difficult to isolate and attribute model differences.

We did not expect the models to be able to simulate the exact deposition at each ice core location given the large natural variability in local weather and snow patterns, which will be different in the models, uncertainties in estimating ice core volcanic sulfate deposition and in model inputs (e.g. magnitude and altitude of the volcanic sulfur emission), and, fundamentally, that in the real world there was only one realization of weather. Regarding the model inputs, there are no direct observations of the injection altitude of $SO_2$ from the 1815 Mt. Tambora eruption, and often simulations are initiated with $SO_2$ injected at heights lower than the estimated injection altitude to account for self-lofting as the aerosol forms. Inter-model uncertainty is also initiated as soon as models convert the same input emission to their grids. Here, simulations followed a common protocol, but it may be that to better simulate the eruption of Mt. Tambora, sensitivity to injection height should be explored. Models may also contain inaccuracies due to uncertain physical representations and coarse resolution, and several ice core locations will be represented by the same model grid box. The differing resolutions between the models also means that the number of grid boxes and area that defines each ice sheet differs slightly between the models. Sulfate deposition fluxes have a large spatial variability due to differences in precipitation, the local synoptic conditions at the time of deposition and post-deposition movement through wind (Fisher et al., 1985; Robock and Free, 1995; Wolff et al., 2005). Deposition fluxes can vary by orders of magnitude, even between ice cores that are located close to each other. For example, in a very low-accumulation site in Antarctica (Dome C), Gautier et al. (2016) found that in five cores drilled 1 m apart, two cores missed the Tambora sulfate flux signal completely, which they attributed to snow drift and surface roughness. They reported that the mean flux between these five cores is uncertain by $\sim 30\%$, highlighting the uncertainties in sulfate fluxes reported from single cores at such a low-accumulation site. This appears to be an extreme case, however, and the 1815 Mt. Tambora signal is clearly identifiable in all other Antarctic ice cores (Sigl et al., 2014).

Furthermore, the phase of the QBO at the time of the eruption is unknown, and here we have only used simulations where the $SO_2$ is emitted during the easterly phase. Toohey et al. (2013) also found that deposition to the poles varied as a function of $SO_2$ injection magnitude and season, and their simulations of a Tambora-like eruption showed greater deposition to Greenland than Antarctica. However, the volcanic eruptions in Toohey et al. (2013) were simulated in January and July, located at $15°$ N, which may explain the bias towards Greenland deposition. Further work is required on the influence of the QBO phase and injection height on the Antarctic and Greenland deposition efficiency.

Our multi-model mean NH_BTD factor is $\sim 60\%$ lower than previously derived (Gao et al., 2007), which if used to estimate the NH atmospheric sulfate burden of other historic tropical eruptions from their mean Greenland de-

posited sulfate would result in lower burdens and likely less volcanic cooling. Model-simulated NH cooling following large-magnitude volcanic eruptions has been overestimated in the past (e.g. Stoffel et al., 2015; Zanchettin et al., 2016). However, our multi-model mean SH_BTD factor is ∼ 30 % greater than Gao et al. (2007), which would result in a larger SH sulfate burden estimate. Applying our BTD factors (NH: 0.42; SH: 1.27) to the mean Greenland and Antarctic deposited sulfate from the 1257 Samalas eruption (90 and 73 kg km$^{-2}$, respectively; Sigl et al., 2015) results in a considerable hemispheric asymmetry in the estimated sulfate burdens. We calculate an SH burden as ∼ 2.5 times the NH burden, despite the eruption occurring in the tropics. This could result in further differences in aerosol optical depth and volcanic aerosol radiative forcing, and hemispheric asymmetry in atmospheric sulfate burdens has been shown to shift the Intertropical Convergence Zone, leading to precipitation anomalies (e.g. Haywood et al., 2013). However, this asymmetry seems unlikely, given that cooling in the SH after large tropical eruptions appears limited in proxy records (Neukom et al., 2014 TS8).

## 5 Conclusions

We have analysed the volcanic sulfate deposition in model simulations of the 1815 eruption of Mt. Tambora using four state-of-the-art global aerosol models (CESM1(WACCM), MAECHAM5-HAM, SOCOL-AER and UM-UKCA) and compared the simulated deposited sulfate to a comprehensive array of ice core records. We have also investigated the simulated sulfate deposition under preindustrial background conditions (without the eruption of Mt. Tambora). Although the models simulate relatively similar background sulfate deposition fluxes, the models differ substantially in their simulation of the Mt. Tambora volcanic sulfate deposition, with differences in the timing, spatial pattern and magnitude. CESM1(WACCM) and UM-UKCA simulate similar deposition patterns, with the majority of sulfate deposited at the midlatitude storm belts. On the ice sheets, UM-UKCA simulates too little deposited sulfate compared to mean ice-core-derived estimates (∼ 2 times too small). In CESM1(WACCM) deposited sulfate on Antarctica is also slightly too small, but deposited sulfate on Greenland is ∼ 2 times too large compared to the mean ice-core-derived estimate. In MAECHAM5-HAM and SOCOL-AER, the sulfate is deposited further across the globe and these models simulate ∼ 3–5 times too much deposited sulfate on the ice sheets compared to mean ice-core-derived estimates. However, MAECHAM5-HAM is the only model to capture the spatial pattern of deposited sulfate compared to ice cores, especially in Antarctica.

Because the background deposition is more comparable between the models than in the perturbed case, differences in the volcanic sulfate deposition are likely due to differences in the formation of the volcanic aerosol, the stratospheric transport of volcanic aerosol and stratosphere–troposphere exchange. In addition, differences in deposition due the deposition schemes may become more pronounced under the higher sulfate loading. We suggest that differences in model resolution, modelled stratospheric winds, aerosol microphysics and sedimentation and deposition schemes have all contributed to the range in model-simulated volcanic sulfate deposition.

We have calculated BTD factors between the mean deposited sulfate on each ice sheet and the corresponding hemispheric peak atmospheric sulfate burden for the Mt. Tambora simulations. The BTD factors differ by up to a factor of 15 between the models. The multi-model mean BTD factors also differ to BTD factors currently used to deduce historical volcanic forcing (e.g. Gao et al., 2007, 2008; Sigl et al., 2015). Our range in derived BTD factors highlights uncertainties in the relationship between atmospheric sulfate burden and ice sheet deposited sulfate as simulated by models.

Given that GISS ModelE (Gao et al., 2007) did as good a job at simulating the deposited sulfate from this eruption as these newer, higher-resolution models, which also have more sophisticated treatments of gas-to-aerosol conversion, and the fact that the four models used here simulate very different sulfate deposition, it remains an open research question as to the optimal model configuration for this problem. A detailed analysis of the differences in sulfur chemistry and the aerosol formation and transport in each model will further aid in the interpretation of these results. Dedicated multi-model comparison projects with process-oriented comparisons, such as the Interactive Stratospheric Aerosol Modelling Intercomparison Project (ISA-MIP) (Timmreck et al., 2016), will be imperative to disentangling the reasons for model differences. Using idealized prescribed aerosol forcings such as Easy Volcanic Aerosol (Toohey et al., 2016) in future VolMIP experiments will also provide the opportunity to better understand model diversity. Simulations of other large-magnitude volcanic eruptions will also enable the calculation of additional multi-model BTD factors, which will aid in the calculation of historic volcanic forcing.

*Data availability.* . TS9

**The Supplement related to this article is available online at https://doi.org/10.5194/acp-18-1-2018-supplement.**

*Competing interests.* The authors declare that they have no conflict of interest.

*Special issue statement.* This article is part of the special issue "The Model Intercomparison Project on the climatic response to

Volcanic forcing (VolMIP) (ESD/GMD/ACP/CP inter-journal SI)". It is not associated with a conference.

*Acknowledgements.* Lauren Marshall is supported by the Natural Environment Research Council (NERC), UK, through the Leeds–York NERC Doctoral Training Partnership. Anja Schmidt was funded by an Academic Research Fellowship from the University of Leeds. Alan Robock is supported by US National Science Foundation grant AGS-1430051. Claudia Timmreck received funding from the German Federal Ministry of Education and Research (BMBF), research program "MiKliP" (FKZ: 01LP1517B). Ulrike Niemeier, Claudia Timmreck and Slimane Bekki acknowledge support from the European Union FP7 project "STRATOCLIM" (FP7-ENV.2013.6.1-2; project 603557). MAECHAM5-HAM simulations were performed at the German climate Computer Centre (DKRZ). Fiona Tummon was funded by Swiss National Science Foundation grant 20F121_138017. This study benefited from the support of the Labex L-IPSL, which is funded by the ANR (grant no. ANR-10-LABX-0018). Matthew Toohey acknowledges support by the Deutsche Forschungsgemeinschaft (DFG) in the framework of the priority programme "Antarctic Research with comparative investigations in Arctic ice areas" through grant TO 967/1-1. Eugene Rozanov and Timofei Sukhodolov acknowledge support from the Swiss National Science Foundation under grant 200021_169241 (VEC). James Pope was funded by NERC grant NEK/K012150/1. The National Center for Atmospheric Research is funded by the National Science Foundation. We thank Chaochao Gao and an anonymous reviewer for their comments and insight on the manuscript, which greatly helped to improve the paper.

Edited by: Ben Kravitz
Reviewed by: Chaochao Gao and one anonymous referee

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
