# Peer review of "Multi-model comparison of the volcanic sulfate deposition from the 1815 eruption of Mt. Tambora"

_Atmospheric Chemistry and Physics, 2017_

## Referee Comment (RC1) · C. Gao (Referee) · 1 Sep 2017

This work is an important contribution to intermodel comparison and evaluation of volcanic simulation. The model derived relationship between volcanic sulfur injection and ice core volcanic aerosol deposition, if the results converge, can provide critical information to verify or improve the ice-core-based reconstruction of past volcanism. The paper is well written; the results are clearly presented and discussed. I would like to recommend the paper to be published in this journal after addressing the following points:

1. In section 2 "Model set-up and ice core data"

Are the four models the only aerosol models available for Tambora simulation? If that's

the case, please state; if not, please explain if there is any criteria taken to choose the models. Also, please briefly describe model performance in previous studies.

2. In section 3 "Results"

2.1 The argument of "models have similar background sulfate deposition patterns" sounds weak to me, please provide more quantitative evidence/analysis to support the argument. The color scale in Figure 1 and few other figures seems misleading, please consider using a monotone color scale.

2.2 A short description of model configuration, especially those closely related to transport dynamic would be helpful to understand the difference in the results. A short summary of the model performance and its implication at the end of this section would be nice.

2.3 P7L3-4, "the four models simulate similar background sulfate deposition patterns and magnitudes and compare well to pre-industrial ice core sulfate fluxes", please provide a table lists all pairs of the model-ice core values to support this statement.

2.4 The differences between wet and dry deposition across models have been discussed at various part of the paper, please explain in more depth what are the implication of these differences.

2.5 Some of the discussions are overlapping or repeating, for example, the temporal evolution of different models in the last paragraph of section 3.2.1 and section 3.2.2. Please consider combine them to shorten the discussion.

3. In section 4 "Discussion"

3.1 Again, some of the discussions repeat the results in section 3. Please focus more on discussing the implication of the results, for example, the causes of the difference in model simulated Tambora deposition.

3.2 Please discuss why models can not give a converged simulation of Tambora deposition, while they were able to simulate the preindustrial background sulfate deposition well.

3.3 P14L28-P15L2, the discussion in these lines seems unnecessary to me since the focus of this study is on model intercomparison.

4. In section 5 "Conclusions"

4.1 P16 L10-11, "Our derived BTD factors highlight uncertainties ..." Not necessary the actual uncertainties between the atmospheric burden and ice sheet deposition, but the uncertainties in the model's ability to derive the relationship. Therefore , I would recommend the authors to rephrase the sentence to make this distinguish.

4.2 P16L19-21, "Using...... will provide the opportunity to better understand model diversity and to advance our understanding of the climate response to large volcanic eruptions". It is true that using the same prescribed forcings could help us to better understand model diversity, but only true to advance the understanding of the volcanic climate response if the prescribed forcings are assumed to be correct. And if that is the case, what about the goal of this VolMIP study to improve the ice-core-based reconstruction?

---

## Referee Comment (RC2) · Anonymous Referee #2 · 21 Sep 2017

Review of Marshall et al. "Multi-model comparison of the volcanic sulfate deposition from the eruption of Mt. Tambora"

This study does a careful comparison of the sulfate aerosol deposition from the Mt. Tambora eruption to test the models ability to simulate deposition observed in the ice core record as well as the assumption made to back out SO2 injections from ice core sulfate signals. This work is done using a variety of models that include microphysical aerosol modules and highlights some of the successes as well as the continued work that needs to be done. I do find that it is a clearly written paper with results that would be of interest to the ACP community and would recommend publication with only a few mostly minor comments for authors to address.

The main comment I have would be related to needing some additional discussion of

hydroxyl radical (OH). I would like the authors to include more details on OH in the main paper and possibly consider a figure in the main text or supplement showing something like profiles of its tropical concentrations in the various models (both background and perturbed if applicable). The table S1 would be best to include in the actual paper rather then the supplement. 3 Models have interactive OH and 1 prescribed. Given OH's critical role in the conversion of SO2 into sulfate aerosol and given the differences in sulfate aerosol evolution in the different models it would be really helpful to look at whether any differences in the amount of OH or its distribution can explain the differences in sulfate conversion noted on page 8 lines 17-20.

For the models with interactive OH does the sulfate aerosol impact photolysis rates, which would decrease OH formation and slow conversion. Does stratospheric water vapor increase in these runs, increasing OH production? Do any of the models deplete OH when reacting with SO2?

If so it would be important to note in the text, if not mention as a source of uncertainty in the sulfate conversion.

page 5 line 2 you should add "as emitting" after simulated or something similar.

page 6 line 20-22 Is there a notable difference between UKCA and other models in this regard that would be worth discussing it seems like a potentially important point concerning the focus of this paper w.r.t. deposition schemes.

In general given the focus of this paper a brief mention of the deposition scheme used in each model and reference would be very helpful.

page 13 lines 13-14 More discussion about OH here and earlier would be helpful

page 13 lines 15-17 I don't think it is beyond the scope of this paper to show or discuss the OH since it is critical in the formation of the sulfate aerosols and could help address difference in the the sulfate burdens

page 14 lines 19-20 sentence starting with "Even if the models were perfect..." I would

recommend removing this sentence, it is not necessary and confusing

Table S1 SOCOL is listed as 8S location of injection the rest are equator is the a typo or real difference in injection latitude.

---

## Author Comment (AC1) · 14 Dec 2017

**Authors' reply to reviewers' comments on 'Multi-model comparison of the volcanic sulfate deposition from the 1815 eruption of Mt. Tambora' by Marshall et al.**

We thank the Reviewers for their detailed and constructive assessment of our manuscript. Their comments and suggestions have greatly helped to improve our paper. Our reply begins with some general changes we have made to the paper, followed by our point-by-point replies. Reviewer comments are in italics and coloured dark grey.

**General changes**

In addition to the reviewer comments, upon revision, the authors have also made the following corrections (see also tracked changed document for minor corrections):

A spin-up issue was discovered in the SOCOL-AER runs and these have since been repeated. In the new runs the carbonyl sulphide (OCS) level was also set to 337 pptv as opposed to ~500 pptv (Table 1). As a consequence, all figures in the manuscript have been updated. Overall the main results in comparison to the other models have not changed, but sulfate deposition over the ice sheets has increased for both background and Tambora deposition. The text has been updated accordingly.

In the revised manuscript we have used the full model names instead of abbreviations to avoid confusion with the use of sub-models as names.

We have rephrased the text under Table 2 regarding the injection details for each model. Despite a uniform injection in CESM1(WACCM) and UM-UKCA, both these injections were divided amongst several grid boxes and each model grid did this differently. As such we thought it was misleading to state that for CESM1(WACCM) the overlap of model levels resulted in the emission fluxes peaking in the centre of the plume and to say nothing of UM-UKCA, where similar effects may be occurring. Instead we have added the sentence: "as the models are not on regular grids and their vertical resolutions differ, the distribution of the emission over the model grid boxes cannot be exactly the same. As a result, the injection profiles differed slightly between the models"

We have also added an additional author, James Pope, who was invaluable in setting up the UM-UKCA runs. We apologise for the oversight in not including him.

**Point-by-point reply**

**Reviewer 1: C. Gao**

*This work is an important contribution to intermodel comparison and evaluation of volcanic simulation. The model derived relationship between volcanic sulfur injection and ice core volcanic aerosol deposition, if the results converge, can provide critical information to verify or improve the ice-core-based reconstruction of past volcanism. The paper is well written; the results are clearly presented and discussed. I would like to recommend the paper to be published in this journal after addressing the following points:*

We thank Chaochao Gao for the helpful comments and have addressed these below.

*1. In section 2 "Model set-up and ice core data". Are the four models the only aerosol models available for Tambora simulation? If that's the case, please state; if not, please explain if there is any criteria taken to choose the models.*

There were no selection criteria as the four models are the only aerosol models available that were able to simulate deposition. A fifth model (CAMB-UPMC-M2D) was included in the initial Tambora experiment as part of Zanchettin et al. (2016), but the model did not simulate deposition and was therefore excluded from the analysis in our paper. We have added this statement to Sect. 2 and have also updated the abstract and introduction to specify that five models simulated the eruption as part of the VolMIP pre-study.

*Also, please briefly describe model performance in previous studies.*

We have added the following paragraph to describe model performance when simulating the eruption of Mt. Pinatubo in 1991:

"All four models simulate the 1991 eruption of Mt. Pinatubo in reasonable agreement with observations of the sulfate burden, aerosol optical depth and stratospheric heating (Niemeier et

al., 2009; Toohey et al., 2011; Dhomse et al., 2014; Sheng et al., 2015b; Mills et al., 2016), giving confidence in the models' overall abilities to accurately simulate the atmospheric and climatic effects of a large-magnitude eruption. However, the models vary in the details regarding the model-observation comparisons. For example, MAECHAM5-HAM (Niemeier et al., 2009) and SOCOL-AER (Sheng et al., 2015b) simulated a too rapid aerosol decay and UM-UKCA (Dhomse et al., 2014) had a low bias in the model-simulated aerosol effective radius compared to observations. Possible reasons for these differences include omitted or under-represented influences from meteoric particles, too large sedimentation and cross-tropopause transport and too fast transport from tropics to high latitudes. Conversely, the models differ in the amount of emitted $SO_2$ required to achieve good comparisons to observations with the mass of $SO_2$ emitted by the four models ranging from 10 Tg for UM-UKCA (Dhomse et al., 2014) and CESM1(WACCM) (Mills et al., 2016; 2017) to 12-14 Tg for SOCOL-AER (Sheng et al., 2015b) to 17 Tg for MAECHAM5-HAM (Niemeier et al., 2009; Toohey et al., 2011). For this reason, the use of a common protocol in this study (Sect. 2.2) enables us to better attribute potential differences in the results to model processes rather than to the eruption source parameters."

*2. In section 3 "Results"*

*2.1 The argument of "models have similar background sulfate deposition  patterns"*
*sounds weak to me, please provide more quantitative evidence/analysis  to  support*
*the argument.  The color scale in Figure 1 and few other figures seems misleading,*
*please consider using a monotone color scale.*

We included the sentence as a statement with the aim of highlighting what can be seen in Figure 1; that the patterns of deposition are similar across the models as described in the opening sentences, despite differences in the magnitude. To make things clearer we have referred to Figure 1. To show quantitatively that the background deposition is similar at the poles we have also moved Figure S4, which shows the ice core fluxes versus the model-simulated values, from the supplementary material to the main text. This is now Figure 3 (please also see reply to point 2.3).

We have considered alternative colour scales but find that the use of this diverging scale highlights the regional variations in deposition more clearly. We therefore retain the original colour scale.

*2.2 A short description of model configuration, especially those closely related to transport dynamic would be helpful to understand the difference in the results. A short summary of the model performance and its implication at the end of this section would be nice.*

We have extended section 2 to include a more detailed paragraph at the beginning to describe the models, with additional section sub-headings. The models all include parameterizations of key aerosol processes such as nucleation, condensation and coagulation and simulate the transport of aerosol through sedimentation and large-scale circulation by the Brewer-Dobson circulation. There are differences in these parameterizations across the models. However, the authors feel that it is beyond the scope of this paper to address these in detail, given that it would be impossible with these simulations alone to determine the relative importance of each process, in addition to differences arising from model resolution and the individual deposition schemes. The inter-model differences found in this study are motivation for more dedicated multi-model comparison projects, which explicitly look at differences in model processes, such as the Interactive Stratospheric Aerosol Modelling Intercomparison Project (ISA-MIP). We have introduced this project in the conclusion.

We have added a brief description of the processes that are included in each model as well as highlighting in this section that the representation of the QBO differs in each model, which will impact the initial transport of sulfate aerosol. We have also added a brief description of the deposition schemes used in the models and further references for each model, which the reader can explore. We have also moved the details from Table 1 in the supplementary to Table 1 in the main paper.

The deposition schemes in SOCOL-AER are relatively simple and are not related to the precipitation. We have therefore removed references to precipitation-deposition correlations for SOCOL-AER throughout the paper.

At the end of section 3.1 we have extended the last paragraph to explain each model's polar deposition and have added the following text to summarize the model performance and the implications:

"Overall, the magnitude of the deposited sulfate in CESM1(WACCM) and MAECHAM5-HAM, where deposition to the ice sheets is dominated by wet deposition, is expected to be

driven by the snow accumulation rates across the ice sheets, which are well represented by all models (Fig. S3). In UM-UKCA, although the polar deposition is correlated with the polar precipitation, the ice sheet sulfate deposition mostly occurs by dry deposition. This is because this model de-activates nucleation scavenging if more than a threshold fraction of the cloud water is present as ice, greatly reducing the aerosol scavenging in polar regions. In SOCOL-AER, fewer regional details are captured since the deposition scheme is simpler and is not connected to precipitation, and therefore the deposition mostly reflects the tropospheric distribution of sulfate.

In summary the models simulate similar overall patterns of background sulfate deposition fluxes, although there are differences in the regional details and magnitude. The similarities and realistic deposition patterns amongst the models suggests that the background sulfate emissions, transport and deposition processes are reasonably parameterized. Although SOCOL-AER is less able to simulate regional details, its simplified deposition scheme is still sufficient for the analysis of inter-hemispheric differences and the temporal evolution of deposition.

*2.3 P7L3-4, "the four models simulate similar background sulfate deposition patterns and magnitudes and compare well to pre-industrial ice core sulfate fluxes", please provide a table lists all pairs of the model-ice core values to support this statement.*

We agree that a table of the comparisons would be useful, and indeed this information was provided in supplementary Figure 4, which showed scatter plots of the ice core fluxes versus those simulated in each model. We have therefore moved this figure to the main text and added a 1:1 line so that the reader can more easily see how each model-simulated value compares to each ice core for both Antarctica and Greenland, and also to the other models. As such, all of the comparisons can be seen in this one figure and we do not think it is necessary to have a table as well. This figure can now be directly compared to the same figure as for the Tambora sulfate ice core deposition fluxes (Figure 7) so the reader can more easily compare how the models perform under the background and perturbed conditions.

The new SOCOL-AER runs show higher deposition to the ice sheets and therefore we have rephrased this sentence to not include SOCOL-AER, where the comparisons are worse than for the other three models. We have also added a sentence to reference the new figure (now Figure 3).

*2.4 The differences between wet and dry deposition across models have been discussed at various part of the paper, please explain in more depth what are the implication of these differences.*

The differences between wet and dry deposition across the models are due to the individual deposition scheme parameterizations. If dry deposition is more spatially uniform across the ice sheets than wet deposition, then the spatial pattern of sulfate deposition flux is a function of the wet deposition, but the total magnitude, dependent on the proportion of wet vs dry deposition. In comparison to differences between sedimentation, aerosol transport and stratosphere-troposphere exchange between the models, we do not think that differences in the wet and dry partitioning is important in explaining these results.

To address this uncertainty, however, we have added the following text to the discussion (Sect. 4.1.3):

"The differences between wet and dry deposition simulated across the models are due to the individual deposition scheme parameterizations. The implication of these differences in dictating the resulting total sulfate deposition remains uncertain. However, since inter-model differences in volcanic sulfate deposition patterns appear unrelated to differences between climatological wet and dry deposition patterns, the proportion of wet vs dry deposition is likely of secondary importance compared to differences between the models in aerosol transport processes including sedimentation and stratosphere-troposphere exchange."

*2.5 Some of the discussions are overlapping or repeating, for example, the temporal evolution of different models in the last paragraph of section 3.2.1 and section 3.2.2.*
*Please consider combine them to shorten the discussion.*

We thank the reviewer for highlighting this and have rearranged section 3.2 and section 3.3 to avoid any major repetitions. We have first focussed on the global sulfate deposition (both spatial and temporal) (section 3.2.1) and then the ice sheet deposition (both spatial and temporal) (3.2.2). We have moved the description of the temporal evolution of sulfate burdens to section 3.3, which is now a dedicated section for all results relating to sulfate burdens and the relationship between burden and deposition. We have renamed the sections accordingly.

We have also moved the paragraphs on winds to the discussion (section 4.1.2).

*3. In section 4 "Discussion"*
*3.1 Again, some of the discussions repeat the results in section 3. Please focus more on discussing the implication of the results, for example, the causes of the difference in model simulated Tambora deposition.*

We have re-written the discussion section to focus more on explaining the differences in model simulated deposition for Tambora and have reduced the emphasis on precipitation (much of this text has been removed). We have split the discussion into the following sections to address each reason (sulfate formation and transport, stratospheric winds and polar vortices, and the deposition schemes) in turn:

4.1 Differences in deposited sulfate

4.1.1 Volcanic sulfate formation and transport

4.1.2 Dynamical effects

4.1.3 Deposition schemes

4.2 Implications for model differences in simulated deposition

We have also removed the sentence referring to aerosol size in different modes (page 13, lines 10-13) since this may be misleading without a comprehensive investigation of aerosol particle size.

*3.2 Please discuss why models cannot give a converged simulation of Tambora deposition, while they were able to simulate the preindustrial background sulfate deposition well.*

Because the models are able to simulate the background deposition reasonably well, the inter-model differences in volcanic sulfate deposition are most likely due to differences in the volcanic aerosol formation and aerosol size due to differences in aerosol microphysics, stratospheric aerosol transport and stratosphere-troposphere exchange. In the background most of the deposited sulfate is of tropospheric origin. Differences in deposition may also become more pronounced in the perturbed case than in the background due to the higher sulfate aerosol burden. Scavenging and deposition parameterizations are highly uncertain, and the chance that

such parameterizations become unrealistic under the large sulfate aerosol loadings associated with a Tambora eruption cannot be discounted, and should be explored in future work.

We hope that the re-written discussion emphasizes these reasons. We have also re-written the conclusions to highlight these reasons.

*3.3 P14L28-P15L2, the discussion in these lines seems unnecessary to me since the focus of this study is on model intercomparison.*

Although we agree that this discussion is surplus to the inter-model focus, we argue that this paragraph is still useful to further put the results in context especially if a reader were to use the ice core comparison to infer model skill. We therefore feel it is necessary to report some of the issues with ice core derived sulfate fluxes.

*4. In section 5 "Conclusions"*
*4.1 P16 L10-11, "Our derived BTD factors highlight uncertainties ..." Not necessary the actual uncertainties between the atmospheric burden and ice sheet deposition, but the uncertainties in the model's ability to derive the relationship. Therefore, I would recommend the authors to rephrase the sentence to make this distinguish.*

The authors agree this was misleading and have rephrased the sentence as follows:
"Our range in derived BTD factors highlights uncertainties in the relationship between atmospheric sulfate burden and ice sheet deposited sulfate as simulated by models"

*4.2 P16L19-21, "Using...... will provide the opportunity to better understand model diversity and to advance our understanding of the climate response to large volcanic eruptions". It is true that using the same prescribed forcings could help us to better understand model diversity, but only true to advance the understanding of the volcanic climate response if the prescribed forcings are assumed to be correct. And if that is the case, what about the goal of this VolMIP study to improve the ice-core-based reconstruction?*

Agreed. We have removed the sentence about advancing our understanding.

**Review 2: Anonymous**

*This study does a careful comparison of the sulfate aerosol deposition from the Mt. Tambora eruption to test the models ability to simulate deposition observed in the ice core record as well as the assumption made to back out SO2 injections from ice core sulfate signals. This work is done using a variety of models that include microphysical aerosol modules and highlights some of the successes as well as the continued work that needs to be done. I do find that it is a clearly written paper with results that would be of interest to the ACP community and would recommend publication with only a few mostly minor comments for authors to address.*

We thank Reviewer 2 for their helpful comments and have addressed these below.

*The main comment I have would be related to needing some additional discussion of hydroxyl radical (OH). I would like the authors to include more details on OH in the main paper and possibly consider a figure in the main text or supplement showing something like profiles of its tropical concentrations in the various models (both background and perturbed if applicable). The table S1 would be best to include in the actual paper rather then the supplement. 3 Models have interactive OH and 1 prescribed. Given OH's critical role in the conversion of SO2 into sulfate aerosol and given the differences in sulfate aerosol evolution in the different models it would be really helpful to look at whether any differences in the amount of OH or its distribution can explain the differences in sulfate conversion noted on page 8 lines 17-20.*

The authors acknowledge the importance of OH in dictating the initial aerosol formation but the effects of OH are second order to differences in aerosol sedimentation and large-scale transport when explaining the inter-model deposition. However, we agree that the paper will be enhanced by adding these details.

We have combined the details of Table S1 with Table 1 and have added an additional paragraph in section 2 to describe the models, which also highlights the OH details in each model. Photolysis rates are not impacted by the sulfate aerosol in any of the models and we have added this statement. Columns labelled "injection height" and "location of injection" are not included in the new Table 1. This information is instead specified under Table 2.

We find that the models have similar background concentrations and distributions of stratospheric OH and have added this statement to the discussion (section 4.1.1) and included this figure in the supplementary (Figure S5). We find that in the models with interactive OH the tropical OH is depleted in the first ~2 months after the eruption. We have also added a supplementary figure of the percentage change in tropical OH concentrations in the first 8 months after the eruption (Figure S6). This figure illustrates the average tropical depletion but depletion rates in the actual aerosol plume would be much higher.

We have introduced the depletion of $SO_2$ in section 3.3 and have added the following text to section 4.1.1:

"MAECHAM5-HAM is the only model that has prescribed OH (Table 1). OH may become depleted in dense volcanic clouds by reaction with $SO_2$, affecting the rate of sulfate aerosol formation (Bekki, 1995). The background stratospheric OH concentrations are similar between the models (Fig. S5) but in SOCOL-AER, UM-UKCA and CESM1(WACCM), in the first 2 months after the eruption, stratospheric tropical OH becomes depleted, with ensemble mean peak reductions of between 15-33% (Fig. S6). This reduces the rate of sulfate aerosol formation compared to MAECHAM5-HAM where the $SO_2$ will be more rapidly oxidised, and explains the later peaks in sulfate burdens in these models."

*For the models with interactive OH does the sulfate aerosol impact photolysis rates, which would decrease OH formation and slow conversion. Does stratospheric water vapor increase in these runs, increasing OH production? Do any of the models deplete OH when reacting with SO2? If so it would be important to note in the text, if not mention as a source of uncertainty in the sulfate conversion.*

Photolysis rates are not impacted by the sulfate aerosol in any of the models and we have added this statement to section 2. Because studies have shown that this effect is not as important as reductions in OH due to depletion by $SO_2$ (e.g. Mills et al., 2017, JGR), we do not think it is necessary to add any more discussion here. The models do deplete the OH when reacting with $SO_2$ (see answer above).

Not all the models outputted the stratospheric water vapour, so we have been unable to investigate this in detail. We find that in all models the OH concentration increases after the

initial depletion (due to oxidation of $SO_2$); in SOCOL-AER, where the stratospheric water vapour was available, we find that the stratospheric water vapour does increase synchronously with the OH increase. These findings are shown in the figures below:

[Figure]

**Figure 1:** Percentage change in tropical OH for each model (ensemble mean)

[Figure]

**Figure 2:** Percentage change in stratospheric water vapour in SOCOL-AER (ensemble mean). Stratospheric water vapour does increase similar to the increase in OH (see Fig. 1).

Given that the later increase in OH does not affect the sulfate aerosol burden (since it is now decaying) and the focus of the paper on deposition, we argue that additional details on stratospheric water vapour are beyond the scope of the paper.

*page 5 line 2 you should add "as emitting" after simulated or something similar.*

We have rephrased the sentence as follows: "The eruption was simulated by emitting the $SO_2$ over 24 hours on 1 April"

*page 6 line 20-22 Is there a notable difference between UKCA and other models in this regard that would be worth discussing it seems like a potentially important point concerning the focus of this paper w.r.t. deposition schemes.*

We have found that UM-UKCA de-activates nucleation scavenging in clouds with a lot of ice, which accounts for why deposition is predominantly dry over the ice sheets. We have mentioned this in the text when explaining the deposition differences, i.e., in section 3.1. Please also see the replies to reviewer 1 point 2.2.

*In general, given the focus of this paper a brief mention of the deposition scheme used in each model and reference would be very helpful.*

We agree that these details would be very helpful so have added a description of these schemes in section 2 along with additional references.

*page 13 lines 13-14 More discussion about OH here and earlier would be helpful*

Please see replies to earlier comments and tracked changes in the revised manuscript.

*page 13 lines 15-17 I don't think it is beyond the scope of this paper to show or discuss the OH since it is critical in the formation of the sulfate aerosols and could help address difference in the the sulfate burdens*

Please see replies to earlier comments. We have included an additional figure (Figure S6) to show the changes in OH after the eruption.

*page 14 lines 19-20 sentence starting with "Even if the models were perfect" I would recommend removing this sentence, it is not necessary and confusing*

We agree and have removed the sentence.

*Table S1 SOCOL is listed as 8S location of injection the rest are equator is the a typo or real difference in injection latitude.*

This is real. We argue that since it is close enough to the equator it can still be defined as an equatorial eruption. We do not think there are significant implications of this that warrant further discussion in the paper. Since combining the information from Table S1 to Table 1, this information has been added as a statement under Table 2.

---

## Author Comment (AC2) · 14 Dec 2017

We thank both reviewers for their detailed and helpful comments. Please see the supplement to our response to reviewer 1 for our responses to both reviewers.

---

## Author Comment (AC4) · 14 Dec 2017

**Supplementary material**

**Table S1:** Ice cores used for volcanic sulfate deposition fluxes after the 1815 eruption of Mt. Tambora and their metadata. Antarctica ice core details taken from Table S1, Sigl et al. (2014).

| Antarctica ice cores | | | | Greenland ice cores | | | |
|---|---|---|---|---|---|---|---|
| **Ice core** | **LAT** | **LON** | **Ref.** | **Ice core** | **LAT** | **LON** | **Ref.** |
| WDC06A | -79.47 | -112.09 | *Sigl et al. (2013)* | B20 | 79 | -36.5 | *Bigler et al. (2002), Gao et al. (2006)* |
| WDC05Q | -79.47 | -112.08 | *Sigl et al. (2013)* | GISP2 | 72.6 | -38.5 | *Gao et al. (2006), Zielinski et al. (1994)* |
| SP04 | -89.95 | 17.67 | *Budner & Cole-Dai, (2003)* | 20D | 65 | -45 | *Gao et al. (2006), Mayewski et al. (1990)* |
| SP01 | -89.95 | 17.67 | *Ferris et al. (2011)* | NGRIP | 75.1 | -42.3 | *Plummer et al. (2012)* |
| DML05 | -75.00 | 0.02 | *Traufetter et al. (2004)* | NEEM-2011-S1 | 77.45 | -51.06 | *Sigl et al. (2013)* |
| DML07 | -75.58 | 3.43 | *Traufetter et al. (2004)* | Humboldt | 78.53 | -56.83 | *Sigl et al. (2013)* |
| B40 | -75.00 | 0.06 | *Sigl et al. (2014)* | Site T | 72.58 | -38.45 | *Mosley-Thompson et al. (2003)* |
| NUS08-4 | -82.82 | 19.90 | *Sigl et al. (2014)* | GITS | 77.14 | -61.095 | *Mosley-Thompson et al. (2003)* |
| NUS08-5 | -82.63 | 17.87 | *Sigl et al. (2014)* | D2 | 71.75 | -46.33 | *Mosley-Thompson et al. (2003)* |
| NUS07-2 | -76.07 | 22.47 | *Sigl et al. (2014)* | D3 | 69.8 | -44.00 | *Mosley-Thompson et al. (2003)* |
| NUS07-5 | -78.65 | 35.63 | *Sigl et al. (2014)* | Raven | 65.9 | -46.3 | *Mosley-Thompson et al. (2003)* |
| NUS07-7 | -82.07 | 54.88 | *Sigl et al. (2014)* | Dye 3 | 65.18 | -43.83 | *Larsen et al. (2008)* |
| EDC96 | -75.10 | 123.35 | *Castellano et al. (2005)* | GRIP | 72.58 | -37.64 | *Larsen et al. (2008)* |
| DFS10 | -77.40 | 39.62 | *Sigl et al. (2014)* | SU07 | 72.5 | -38.5 | *Cole-Dai et al. (2009)* |
| DF01 | -77.37 | 39.70 | *Motizuki et al. (2014)* | | | | |
| W10k | -66.75 | 112.83 | *Sigl et al. (2014)* | | | | |
| DIV2010 | -77.95 | -95.96 | *Sigl et al. (2014)* | | | | |
| NUS08-7 | -74.88 | 1.60 | *Sigl et al. (2014)* | | | | |
| NUS07-1 | -73.72 | 7.98 | *Sigl et al. (2014)* | | | | |
| TalosDome | -72.48 | 159.06 | *Stenni et al. (2002)* | | | | |
| Taylor Dome | -77.81 | 158.72 | *Mayewski et al. (1996)* | | | | |
| DomeA | -80.37 | 77.22 | *Jiang et al. (2012)* | | | | |
| DSS | -66.77 | 112.80 | *Plummer et al. (2012)* | | | | |
| Siple | -75.91 | -83.91 | *Cole-Dai et al. (1997)* | | | | |
| Dyer | -70.66 | -64.87 | *Cole-Dai et al. (1997)* | | | | |
| PlatRemote | -84.00 | 43.00 | *Cole-Dai et al. (2000)* | | | | |

**Table S2:** Ice cores used for pre-industrial background sulfate deposition fluxes (1850-1860 mean) taken from Lamarque et al. (2013).

| Antarctic ice cores | | | Arctic ice cores | | |
|---|---|---|---|---|---|
| **Ice core** | **LAT** | **LON** | **Ice core** | **LAT** | **LON** |
| W10 | -66.3 | 112.83 | ACT11d | 66.47 | -46.3 |
| DIV | -76.77 | -101.73 | D4 | 71.4 | -43.9 |
| WD | -79.47 | -112.68 | Zoe | 72.6 | -38.3 |
| NUS Site8_7 | -74.88 | 1.6 | NEEMS3 | 77.43 | -51.05 |
| NUS Site8_5 | -82.63 | 17.87 | Tunu | 78.00 | -33.98 |
| NUS Site7_7 | -82.07 | 54.88 | McCall | 69.3 | -143.8 |
| NUS Site7_5 | -78.65 | 35.63 | Akademmi Nauk | 80.52 | 94.82 |
| NUS Site7_2 | -76.07 | 22.47 | Flade Isblink | 81.58 | -15.7 |
| NUS Site7_1 | -73.72 | 7.98 | | | |

**Table S3:** Mean polar (60°-90°) cumulative deposited sulfate [kg $SO_4$ km$^{-2}$] and revised BTD factors [$* 10^9$ km$^{-2}$] calculated from mean polar deposited sulfate and hemispheric peak atmospheric sulfate burden as opposed to ice sheet deposited sulfate (ensemble mean).

| Model | Arctic deposition [kg $SO_4$ km$^{-2}$] | NH_BTD [$10^9$ km$^{-2}$] | Antarctic deposition [kg $SO_4$ km$^{-2}$] | SH_BTD [$10^9$ km$^{-2}$] |
|---|---|---|---|---|
| CESM1(WACCM) | 125 | 0.27 | 100 | 0.58 |
| MAECHAM5-HAM | 175 | 0.21 | 287 | 0.17 |
| SOCOL-AER | 131 | 0.25 | 168 | 0.33 |
| UM-UKCA | 77 | 0.38 | 53 | 1.07 |

[Figure]

**Figure S1:** Pre-industrial background (no Tambora) global atmospheric sulfate burdens in the model control simulations (year average).

[Figure]

**Figure S2:** Pre-industrial background global precipitation in each model control simulation (year average). SOCOL-AER is included here for reference but deposition in SOCOL-AER is not connected to the precipitation.

[Figure]

**Figure S3:** Pre-industrial background polar precipitation in each model control simulation (year average) (shading) and ice core accumulation (mm liquid water equivalent yr$^{-1}$) in ice cores (filled circles) (Sigl et al., 2014). Antarctic ice core accumulation rates are an average of annual ice core accumulation from 1850-1860 taken from Sigl et al. (2014). Greenland ice core accumulation rates are taken from Gao et al. (2006) (their Table 1). SOCOL-AER is included here for reference but deposition in SOCOL-AER is not connected to the precipitation.

[Figure]

**Figure S4:** Cumulative deposited sulfate [kg $SO_4$ km$^{-2}$] for MAECHAM5-HAM and SOCOL-AER (ensemble mean). Results have been reduced by a factor of 3 (for MAECHAM5-HAM the slope of the regression line between simulated deposited sulfate and ice core records in Antarctica was 3.7 and 1.7 in Greenland. SOCOL-AER is reduced by the same factor for comparison). MAECHAM5-HAM is able to simulate the spatial pattern of ice sheet deposited sulfate when compared to ice cores, but the magnitude is too large.

[Figure]

**Figure S5:** Annual-mean zonal-mean OH [ppmv] in each model's pre-industrial background control simulations. In MAECHAM5-HAM the OH is prescribed.

[Figure]

**Figure S6:** Percentage change in tropical (15°S - 15°N) OH in the first 8 months after the eruption (ensemble mean) for each model including interactive OH chemistry.

[Figure]

**Figure S7:** Tropical mean (15°S - 15°N) zonal wind for the volcanic simulations in each model (ensemble mean). Tropical winds in UM-UKCA, SOCOL-AER and CESM1(WACCM) oscillate, exhibiting characteristics of the QBO, with downward propagating easterly and westerly winds, but length of phase differs. QBO easterly phase is longer in UM-UKCA; ~2.5 years compared to ~1.5 years in CESM1(WACCM) and SOCOL-AER. MAECHAM5-HAM does not include representation of the QBO and winds remain easterly in the lower stratosphere throughout the simulations.

[revised manuscript text omitted]